# Label Noise SGD Provably Prefers Flat Global Minimizers

**Alex Damian**
Princeton University
ad27@princeton.edu

**Tengyu Ma**
Stanford University
tengyuma@stanford.edu

**Jason Lee**
Princeton University
jasonlee@princeton.edu

## Abstract

In overparametrized models, the noise in stochastic gradient descent (SGD) implicitly regularizes the optimization trajectory and determines which local minimum SGD converges to. Motivated by empirical studies that demonstrate that training with noisy labels improves generalization, we study the implicit regularization effect of SGD with label noise. We show that SGD with label noise converges to a stationary point of a regularized loss $L(\theta) + \lambda R(\theta)$, where $L(\theta)$ is the training loss, $\lambda$ is an effective regularization parameter depending on the step size, strength of the label noise, and the batch size, and $R(\theta)$ is an explicit regularizer that penalizes sharp minimizers. Our analysis uncovers an additional regularization effect of large learning rates beyond the linear scaling rule that penalizes large eigenvalues of the Hessian more than small ones. We also prove extensions to classification with general loss functions, significantly strengthening the prior work of Blanc et al. [3] to global convergence and large learning rates and of HaoChen et al. [12] to general models.

## 1 Introduction

One of the central questions in modern machine learning theory is the generalization capability of overparametrized models trained by stochastic gradient descent (SGD). Recent work identifies the implicit regularization effect due to the optimization algorithm as one key factor in explaining the generalization of overparameterized models [27, 11, 19, 10]. This implicit regularization is controlled by many properties of the optimization algorithm including search direction [11], learning rate [20], batch size [26], momentum [21] and dropout [22].

The parameter-dependent noise distribution in SGD is a crucial source of regularization [16, 18]. Blanc et al. [3] initiated the study of the regularization effect of label noise SGD with square loss[1] by characterizing the local stability of global minimizers of the training loss. By identifying a data-dependent regularizer $R(\theta)$, Blanc et al. [3] proved that label noise SGD locally diverges from the global minimizer $\theta^*$ if and only if $\theta^*$ is not a first-order stationary point of $\min_\theta R(\theta)$ subject to $L(\theta) = 0$. The analysis is only able to demonstrate that with sufficiently small step size $\eta$, label noise SGD initialized at $\theta^*$ locally diverges by a distance of $\eta^{0.4}$ and correspondingly decreases the regularizer by $\eta^{0.4}$. This is among the first results that establish that the noise distribution alters the local stability of stochastic gradient descent. However, the parameter movement of $\eta^{0.4}$ is required to

---

[1] Label noise SGD computes the stochastic gradient by first drawing a sample $(x_i, y_i)$, perturbing $y_i' = y_i + \epsilon$ with $\epsilon \sim \{-\sigma, \sigma\}$, and computing the gradient with respect to $(x_i, y_i')$.

35th Conference on Neural Information Processing Systems (NeurIPS 2021).

be inversely polynomially small in dimension and condition number and is thus too small to affect the predictions of the model.

HaoChen et al. [12], motivated by the local nature of Blanc et al. [3], analyzed label noise SGD in the quadratically-parametrized linear regression model [29, 32, 23]. Under a well-specified sparse linear regression model and with isotropic features, HaoChen et al. [12] proved that label noise SGD recovers the sparse ground-truth despite overparametrization, which demonstrated a global implicit bias towards sparsity in the quadratically-parametrized linear regression model.

This work seeks to identify the global implicit regularization effect of label noise SGD. Our primary result, which supports Blanc et al. [3], proves that label noise SGD converges to a stationary point of $L(\theta) + \lambda R(\theta)$, where the regularizer $R(\theta)$ penalizes sharp regions of the loss landscape.

The focus of this paper is on label noise SGD due to its strong regularization effects in both real and synthetic experiments [25, 28, 31]. Furthermore, label noise is used in large-batch training as an additional regularizer [25] when the regularization from standard regularizers (e.g. mini-batch, batch-norm, and dropout) is not sufficient. Label noise SGD is also known to be less sensitive to initialization, as shown in HaoChen et al. [12]. In stark contrast, mini-batch SGD remains stuck when initialized at any poor global minimizer. Our analysis demonstrates a global regularization effect of label noise SGD by proving it converges to a stationary point of a regularized loss $L(\theta) + \lambda R(\theta)$, even when initialized at a zero error global minimum.

The learning rate and minibatch size in SGD are known to be important sources of regularization [9]. Our main theorem highlights the importance of learning rate and batch size as the hyperparameters that control the balance between the loss and the regularizer – larger learning rates and smaller batch sizes lead to stronger regularization.

Section 2 reviews the notation and assumptions used throughout the paper. Section 2.4 formally states the main result and Section 3 sketches the proof. Section 4 presents experimental results which support our theory. Finally, Section 6 discusses the implications of this work.

## 2 Problem Setup and Main Result

Section 2.1 describes our notation and the SGD with label noise algorithm. Section 2.2 introduces the explicit formula for the regularizer $R(\theta)$. Sections 2.3 and 2.4 formally state our main result.

### 2.1 Notation

We focus on the regression setting (see Appendix E for the extension to the classification setting). Let $\{(x_i, y_i)\}_{i \in [n]}$ be $n$ datapoints with $x_i \in \mathcal{D}$ and $y_i \in \mathbb{R}$. Let $f : \mathcal{D} \times \mathbb{R}^d \to \mathbb{R}$ and let $f_i(\theta) = f(x_i, \theta)$ denote the value of $f$ on the datapoint $x_i$. Define $\ell_i(\theta) = \frac{1}{2} \left( f_i(\theta) - y_i \right)^2$ and $L(\theta) = \frac{1}{n} \sum_{i=1}^n \ell_i(\theta)$. Then we will follow Algorithm 1 which adds fresh additive noise to the labels $y_i$ at every step before computing the gradient:

---
**Algorithm 1:** SGD with Label Noise

---
**Input:** $\theta_0$, step size $\eta$, noise variance $\sigma^2$, batch size $B$, steps $T$
**for** $k = 0$ *to* $T - 1$ **do**
  Sample batch $\mathcal{B}^{(k)} \subset [n]^B$ uniformly and label noise $\epsilon_i^{(k)} \sim \{-\sigma, \sigma\}$ for $i \in \mathcal{B}^{(k)}$.
  Let $\hat{\ell}_i^{(k)}(\theta) = \frac{1}{2} \left( f_i(\theta) - y_i - \epsilon_i^{(k)} \right)^2$ and $\hat{L}^{(k)} = \frac{1}{B} \sum_{i \in \mathcal{B}^{(k)}} \hat{\ell}_i^{(k)}$.
  $\theta_{k+1} \leftarrow \theta_k - \eta \nabla \hat{L}^{(k)}(\theta_k)$
**end**

---

Note that $\sigma$ controls the strength of the label noise and will control the strength of the implicit regularization in Theorem 1. Throughout the paper we will use $\| \cdot \| = \| \cdot \|_2$. We make the following standard assumption on $f$:

**Assumption 1** (Smoothness). *We assume that each $f_i$ is $\ell_f$-Lipschitz, $\nabla f_i$ is $\rho_f$-Lipschitz, and $\nabla^2 f_i$ is $\kappa_f$-Lipschitz with respect to $\| \cdot \|_2$ for $i = 1, \ldots, n$.*

We will define $\ell = \ell_f^2$ to be an upper bound on $\|\frac{1}{n}\sum_i \nabla f_i(\theta)\nabla f_i(\theta)^T\|_2$, which is equal to $\|\nabla^2 L(\theta)\|_2$ at any global minimizer $\theta$. Our results extend to any learning rate $\eta \in (0, \frac{2}{\ell})$. However, they do not extend to the limit as $\eta \to \frac{2}{\ell}$. Because we still want to track the dependence on $\frac{1}{\eta}$, we do not assume $\eta$ is a fixed constant and instead assume some constant separation:

**Assumption 2** (Learning Rate Separation). *There exists a constant $\nu \in (0, 1)$ such that $\eta \le \frac{2-\nu}{\ell}$.*

In addition, we make the following local Kurdyka-Łojasiewicz assumption (KL assumption) which ensures that there are no regions where the loss is very flat. The KL assumption is very general and holds for some $\delta > 0$ for any analytic function defined on a compact domain (see Lemma 17).

**Assumption 3** (KL). *Let $\theta^*$ be any global minimizer of $L$. Then there exist $\epsilon_{KL} > 0, \mu > 0$ and $0 < \delta \le 1/2$ such that if $L(\theta) - L(\theta^*) \le \epsilon_{KL}$, then $L(\theta) - L(\theta^*) \le \mu \|\nabla L(\theta)\|^{1+\delta}$.*

We assume $L(\theta^*) = 0$ for any global minimizer $\theta^*$. Note that if $L$ satisfies Assumption 3 for some $\delta$ then it also satisfies Assumption 3 for any $\delta' < \delta$. Assumption 3 with $\delta = 1$ is equivalent to the much stronger Polyak-Łojasiewicz condition which is equivalent to local strong convexity.

We will use $O, \Theta, \Omega$ to hide any polynomial dependence on $\mu, \ell_f, \rho_f, \kappa_f, \nu, 1/\sigma, n, d$ and $\tilde{O}$ to hide additional polynomial dependence on $\log 1/\eta, \log B$.

## 2.2 The Implicit Regularizer $R(\theta)$

For $L, \sigma^2, B, \eta$ as defined above, we define the implicit regularizer $R(\theta)$, the effective regularization parameter $\lambda$, and the regularized loss $\tilde{L}(\theta)$:

$$R(\theta) = -\frac{1}{2\eta}\operatorname{tr}\log\left(1 - \frac{\eta}{2}\nabla^2 L(\theta)\right), \qquad \lambda = \frac{\eta\sigma^2}{B}, \qquad \tilde{L}(\theta) = L(\theta) + \lambda R(\theta). \quad (1)$$

Here $\log$ refers to the matrix logarithm. To better understand the regularizer $R(\theta)$, let $\lambda_1, \ldots, \lambda_d$ be the eigenvalues of $\nabla^2 L(\theta)$ and let $R(\lambda_i) = -\frac{1}{2\eta}\log(1 - \frac{\eta\lambda_i}{2})$. Then,

$$R(\theta) = \sum_{i=1}^d R(\lambda_i) = \sum_{i=1}^d \left(\frac{\lambda_i}{4} + \frac{\eta\lambda_i^2}{16} + \frac{\eta^2\lambda_i^3}{48} + \ldots\right).$$

In the limit as $\eta \to 0$, $R(\theta) \to \frac{1}{4}\operatorname{tr}\nabla^2 L(\theta)$, which matches the regularizer in Blanc et al. [3] for infinitesimal learning rate near a global minimizer. However, in additional to the linear scaling rule, which is implicit in our definition of $\lambda$, our analysis uncovers an **additional regularization effect of large learning rates** that penalizes larger eigenvalues more than smaller ones (see Figure 1 and Section 6.1).

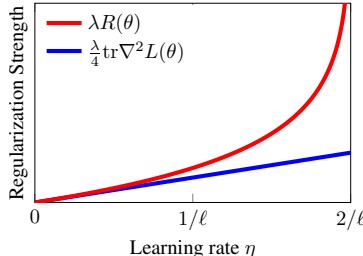

Figure 1: Regularization strength as a function of $\eta$

The goal of this paper is to show that Algorithm 1 converges to a stationary point of the regularized loss $\tilde{L} = L + \lambda R$. In particular, we will show convergence to an $(\epsilon, \gamma)$-stationary point, which is defined in the next section.

## 2.3 $(\epsilon, \gamma)$-Stationary Points

We begin with the standard definition of an approximate stationary point:

**Definition 1** ($\epsilon$-stationary point). *$\theta$ is an $\epsilon$-stationary point of $f$ if $\|\nabla f(\theta)\| \le \epsilon$.*

In stochastic gradient descent it is often necessary to allow $\lambda = \frac{\eta\sigma^2}{B}$ to scale with $\epsilon$ to reach an $\epsilon$-stationary point [8, 15] (e.g., $\lambda$ may need to be less than $\epsilon^2$). However, for $\lambda = O(\epsilon)$, any local minimizer $\theta^*$ is an $\epsilon$-stationary point of $\tilde{L} = L + \lambda R$. Therefore, reaching a $\epsilon$-stationary point of $\tilde{L}$ would be equivalent to finding a local minimizer and would not be evidence for implicit regularization. To address this scaling issue, we consider the rescaled regularized loss:

$$\frac{1}{\lambda}\tilde{L} = \frac{1}{\lambda}L + R.$$

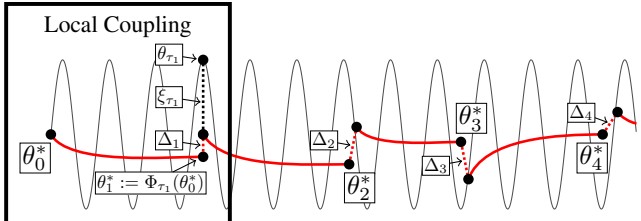

Figure 2: **Local Coupling:** We decompose $\theta$ as the sum of a regularized trajectory $\Phi_{\tau_1}(\theta_0^*)$, a mean zero oscillating process $\xi_{\tau_1}$, and an error term $\Delta_1$. **Global Convergence:** We repeat the coupling with a sequence of reference points $\{\theta_m^*\}_m$ to prove convergence to a stationary point of $\frac{1}{\lambda}\tilde{L}$.

Reaching an $\epsilon$-stationary point of $\frac{1}{\lambda}\tilde{L}$ requires non-trivially taking the regularizer $R$ into account. However, it is not possible for Algorithm 1 to reach an $\epsilon$-stationary point of $\frac{1}{\lambda}\tilde{L}$ even in the ideal setting when $\theta$ is initialized near a global minimizer $\theta^*$ of $\tilde{L}$. The label noise will cause fluctuations of order $\sqrt{\lambda}$ around $\theta^*$ (see section 3) so $\|\nabla L\|$ will remain around $\sqrt{\lambda}$. This causes $\frac{1}{\lambda}\nabla L$ to become unbounded for $\lambda$ (and therefore $\epsilon$) sufficiently small, and thus Algorithm 1 cannot converge to an $\epsilon$-stationary point. We therefore prove convergence to an $(\epsilon, \gamma)$-*stationary point*:

**Definition 2** (($\epsilon, \gamma$)-stationary point). *$\theta$ is an $(\epsilon, \gamma)$-stationary point of $f$ if there exists some $\theta^*$ such that $\|\nabla f(\theta^*)\| \leq \epsilon$ and $\|\theta - \theta^*\| \leq \gamma$.*

Intuitively, Algorithm 1 converges to an $(\epsilon, \gamma)$-stationary point when it converges to a neighborhood of some $\epsilon$-stationary point $\theta^*$.

## 2.4 Main Result

Having defined an $(\epsilon, \gamma)$-stationary point we can now state our main result:

**Theorem 1.** *Assume that $f$ satisfies Assumption 1, $\eta$ satisfies Assumption 2, and $L$ satisfies Assumption 3, i.e. $L(\theta) \leq \mu\|\nabla L(\theta)\|^{1+\delta}$ for $L(\theta) \leq \epsilon_{KL}$. Let $\eta, B$ be chosen such that $\lambda := \frac{\eta\sigma^2}{B} = \tilde{\Theta}(\min(\epsilon^{2/\delta}, \gamma^2))$, and let $T = \tilde{\Theta}(\eta^{-1}\lambda^{-1-\delta}) = \text{poly}(\eta^{-1}, \gamma^{-1})$. Assume that $\theta$ is initialized within $O(\sqrt{\lambda^{1+\delta}})$ of some $\theta^*$ satisfying $L(\theta^*) = O(\lambda^{1+\delta})$. Then for any $\zeta \in (0, 1)$, with probability at least $1 - \zeta$, if $\{\theta_k\}$ follows Algorithm 1 with parameters $\eta, \sigma, T$, there exists $k < T$ such that $\theta_k$ is an $(\epsilon, \gamma)$-stationary point of $\frac{1}{\lambda}\tilde{L}$.*

Theorem 1 guarantees that Algorithm 1 will hit an $(\epsilon, \gamma)$-stationary point of $\frac{1}{\lambda}\tilde{L}$ within a polynomial number of steps in $\epsilon^{-1}, \gamma^{-1}$. In particular, when $\delta = \frac{1}{2}$, Theorem 1 guarantees convergence within $\tilde{O}(\epsilon^{-6} + \gamma^{-3})$ steps. The condition that $\theta_0$ is close to an approximate global minimizer $\theta^*$ is not a strong assumption as recent methods have shown that overparameterized models can easily achieve zero training loss in the kernel regime (see Appendix C). However, in practice these minimizers of the training loss generalize poorly [1]. Theorem 1 shows that Algorithm 1 can then converge to a stationary point of the regularized loss which has better generalization guarantees (see Section 6.2). Theorem 1 also generalizes the local analysis in Blanc et al. [3] to a global result with weaker assumptions on the learning rate $\eta$. For a full comparison with Blanc et al. [3], see section 3.1.

## 3 Proof Sketch

The proof of convergence to an $(\epsilon, \varphi)$-stationary point of $\frac{1}{\lambda}\tilde{L}$ has two components. In Section 3.1, we pick a reference point $\theta^*$ and analyze the behavior of Algorithm 1 in a neighborhood of $\theta^*$. In Section 3.2, we repeat this local analysis with a sequence of reference points $\{\theta_m^*\}$.

### 3.1 Local Coupling

Let $\Phi_k(\cdot)$ denote $k$ steps of gradient descent on the regularized loss $\tilde{L}$, i.e.

$$\Phi_0(\theta) = \theta \qquad \text{and} \qquad \Phi_{k+1}(\theta) = \Phi_k(\theta) - \eta\nabla\tilde{L}(\Phi_k(\theta)), \tag{2}$$

where $\tilde{L}(\theta) = L(\theta) + \lambda R(\theta)$ is the regularized loss defined in Equation (1). Lemma 1 states that if $\theta$ is initialized at an approximate global minimizer $\theta^*$ and follows Algorithm 1, there is a small mean zero random process $\xi$ such that $\theta_k \approx \Phi_k(\theta^*) + \xi_k$:

**Lemma 1.** *Let*

$$\iota = c \log \frac{d}{\lambda \zeta}, \quad \mathscr{X} = \sqrt{\frac{2\lambda n d \iota}{\nu}}, \quad \mathscr{L} = c\lambda^{1+\delta}, \quad \mathscr{D} = c\sqrt{\mathscr{L}}\iota, \quad \mathscr{M} = \frac{\mathscr{D}}{\nu}, \quad \mathscr{T} = \frac{1}{c^2 \eta \mathscr{X} \iota},$$

*where $c$ is a sufficiently large constant. Assume $f$ satisfies Assumption 1 and $\eta$ satisfies Assumption 2. Let $\theta$ follow Algorithm 1 starting at $\theta^*$ and assume that $L(\theta^*) \leq \mathscr{L}$ for some $0 < \delta \leq 1/2$. Then there exists a random process $\{\xi_k\}$ such that for any $\tau \leq \mathscr{T}$ satisfying $\max_{k \leq \tau} \|\Phi_k(\theta^*) - \theta^*\| \leq 8\mathscr{M}$, with probability at least $1 - 10 d\tau e^{-\iota}$ we have simultaneously for all $k \leq \tau$,*

$$\|\theta_k - \xi_k - \Phi_k(\theta^*)\| \leq \mathscr{D}, \qquad \mathbb{E}[\xi_k] = 0, \qquad and \qquad \|\xi_k\| \leq \mathscr{X}.$$

Note that because $\mathscr{M} \geq \mathscr{D}$, the error term $\mathscr{D}$ is at least 8 times smaller than the movement in the direction of the regularized trajectory $\Phi_\tau(\theta^*)$, which will allow us to prove convergence to an $(\epsilon, \gamma)$-stationary point of $\frac{1}{\lambda}\tilde{L}$ in Section 3.2.

Toward simplifying the update in Algorithm 1, we define $L^{(k)}$ to be the true loss without label noise on batch $\mathcal{B}^{(k)}$. The label-noise update $\hat{L}^{(k)}(\theta_k)$ is an unbiased perturbation of the mini-batch update: $\nabla \hat{L}^{(k)}(\theta_k) = \nabla L^{(k)}(\theta_k) - \frac{1}{B}\sum_{i \in \mathcal{B}^{(k)}} \epsilon_i^{(k)} \nabla f_i(\theta_k)$. We decompose the update rule into three parts:

$$\theta_{k+1} = \theta_k - \underbrace{\eta \nabla L(\theta_k)}_{\text{gradient descent}} - \underbrace{\eta[\nabla L^{(k)}(\theta_k) - \nabla L(\theta_k)]}_{\text{minibatch noise}} + \underbrace{\frac{\eta}{B}\sum_{i \in \mathcal{B}^{(k)}} \epsilon_i^{(k)} \nabla f_i(\theta_k)}_{\text{label noise}}. \tag{3}$$

Let $m_k = -\eta[\nabla L^{(k)}(\theta_k) - \nabla L(\theta_k)]$ denote the minibatch noise. Throughout the proof we will show that the minibatch noise is dominated by the label noise. We will also decompose the label noise into two terms. The first, $\epsilon_k^*$, will represent the label noise if the gradient were evaluated at $\theta^*$ whose distribution does not vary with $k$. The other term, $z_k$ represents the change in the noise due to evaluating the gradient at $\theta_k$ rather than $\theta^*$. More precisely, we have

$$\epsilon_k^* = \frac{\eta}{B}\sum_{i \in \mathcal{B}^{(k)}} \epsilon_i^{(k)} \nabla f_i(\theta^*) \qquad and \qquad z_k = \frac{\eta}{B}\sum_{i \in \mathcal{B}^{(k)}} \epsilon_i^{(k)}[\nabla f_i(\theta_k) - \nabla f_i(\theta^*)].$$

We define $G(\theta) = \frac{1}{n}\sum_i \nabla f_i(\theta) \nabla f_i(\theta)^T$ to be the covariance of the model gradients. Note that $\epsilon_k^*$ has covariance $\eta\lambda G(\theta^*)$. To simplify notation in the Taylor expansions, we will use the following shorthand to refer to various quantities evaluated at $\theta^*$:

$$G = G(\theta^*), \qquad \nabla^2 L = \nabla^2 L(\theta^*), \qquad \nabla^3 L = \nabla^3 L(\theta^*), \qquad \nabla R = \nabla R(\theta^*).$$

First we need the following standard decompositions of the Hessian:

**Proposition 1.** *For any $\theta \in \mathbb{R}^d$ we can decompose $\nabla^2 L(\theta) = G(\theta) + E(\theta)$ where $E(\theta) = \frac{1}{n}\sum_{i=1}^n (f_i(\theta) - y_i)\nabla^2 f_i(\theta)$ satisfies $\|E(\theta)\| \leq \sqrt{2\rho_f L(\theta)}$ where $\rho_f$ is defined in Assumption 1.*

The matrix $G$ in Proposition 1 is known as the Gauss-Newton term of the Hessian. We can now Taylor expand Algorithm 1 and Equation (2) to first order around $\theta^*$:

$$\Phi_{k+1}(\theta^*) \approx \Phi_k(\theta^*) - \eta\left[\nabla L + \nabla^2 L(\Phi_k(\theta^*) - \theta^*)\right],$$

$$\theta_{k+1} \approx \theta_k - \eta\left[\nabla L + \nabla^2 L(\theta_k - \theta^*)\right] + \epsilon_k^*.$$

We define $v_k = \theta_k - \Phi_k(\theta^*)$ to be the deviation from the regularized trajectory. Then subtracting these two equations gives

$$v_{k+1} \approx (I - \eta\nabla^2 L)v_k + \epsilon_k^* \approx (I - \eta G)v_k + \epsilon_k^*,$$

where we used Proposition 1 to replace $\nabla^2 L$ with $G$. Temporarily ignoring the higher order terms, we define the random process $\xi$ by

$$\xi_{k+1} = (I - \eta G)\xi_k + \epsilon_k^* \qquad and \qquad \xi_0 = 0. \tag{4}$$

The process $\xi$ is referred to as an Ornstein Uhlenbeck process and it encodes the movement of $\theta$ to first order around $\theta^*$. We defer the proofs of the following properties of $\xi$ to Appendix B:

**Proposition 2.** *For any $k \geq 0$, with probability at least $1 - 2de^{-\iota}$, $\|\xi_k\| \leq \mathscr{X}$. In addition, as $k \to \infty$, $\mathbb{E}[\xi_k \xi_k^T] \to \lambda \Pi_G (2 - \eta G)^{-1}$ where $\Pi_G$ is the projection onto the span of $G$.*

We can now analyze the effect of $\xi_k$ on the second order Taylor expansion. Let $r_k = \theta_k - \Phi_k(\theta^*) - \xi_k$ be the deviation of $\theta$ from the regularized trajectory after removing the Ornstein Uhlenbeck process $\xi$. Lemma 1 is equivalent to $\Pr[\|r_\tau\| \geq \mathscr{D}] \leq 10\tau de^{-\iota}$.

We will prove by induction that $\|r_k\| \leq \mathscr{D}$ for all $k \leq t$ with probability at least $1 - 10td e^{-\iota}$ for all $t \leq \tau$. The base case follows from $r_0 = 0$ so assume the result for some $t \geq 0$. The remainder of this section will be conditioned on the event $\|r_k\| \leq \mathscr{D}$ for all $k \leq t$. $O(\cdot)$ notation will only be used to hide absolute constants that do not change with $t$ and will additionally not hide dependence on the absolute constant $c$. The following proposition fills in the missing second order terms in the Taylor expansion around $\theta^*$ of $r_k$:

**Proposition 3.** *With probability at least $1 - 2de^{-\iota}$,*

$$r_{k+1} = (I - \eta G)r_k - \eta \left[ \frac{1}{2} \nabla^3 L(\xi_k, \xi_k) - \lambda \nabla R \right] + m_k + z_k + \tilde{O}\left( c^{5/2} \eta \lambda^{1+\delta} \right)$$

The intuition for the implicit regularizer $R(\theta)$ is that by Propositions 1 and 2,

$$\mathbb{E}[\xi_k \xi_k^T] \to \Pi_G \lambda (2 - \eta G)^{-1} \approx \lambda (2 - \eta \nabla^2 L)^{-1}.$$

Therefore, when averaged over long timescales,

$$\frac{1}{2} \mathbb{E}[\nabla^3 L(\xi_k, \xi_k)] \approx \frac{\lambda}{2} \nabla^3 L \left[ (2 - \eta \nabla^2 L)^{-1} \right] = \lambda \nabla \left[ -\frac{1}{2\eta} \operatorname{tr} \log \left( 1 - \frac{\eta}{2} \nabla^2 L(\theta) \right) \right]\bigg|_{\theta = \theta^*} = \lambda \nabla R.$$

The second equality follows from the more general equality that for any matrix function $A$ and any scalar function $h$ that acts independently on each eigenvalue, $\nabla(\operatorname{tr} h(A(\theta))) = (\nabla A(\theta))(h'(A(\theta)))$ which follows from the chain rule. The above equality is the special case when $A(\theta) = \nabla^2 L(\theta)$ and $h(x) = -\frac{1}{\eta} \log \left( 1 - \frac{\eta}{2} x \right)$, which satisfies $h'(x) = \frac{1}{2 - \eta x}$.

The remaining details involve concentrating the mean zero error terms $m_k, z_k$ and showing that $\mathbb{E}[\xi_k \xi_k^T]$ *does* concentrate in the directions with large eigenvalues and that the directions with small eigenvalues, in which the covariance does not concentrate, do not contribute much to the error. This yields the following bound:

**Proposition 4.** *With probability at least $1 - 10de^{-\iota}$, $\|r_{t+1}\| = \tilde{O}\left( \frac{\lambda^{1/2+\delta/2}}{\sqrt{c}} \right)$.*

The proof of Proposition 4 can be found in Appendix B. Finally, because $\mathscr{D} = \tilde{O}(c^{5/2} \lambda^{1/2+\delta/2})$, $\|r_{t+1}\| \leq \mathscr{D}$ for sufficiently large $c$. This completes the induction and the proof of Lemma 1.

**Comparison with Blanc et al. [3]** Like Blanc et al. [3], Lemma 1 shows that $\theta$ locally follows the trajectory of gradient descent on an implicit regularizer $R(\theta)$. However, there are a few crucial differences:

- Because we do not assume we start near a global minimizer where $L = 0$, we couple to a regularized loss $\tilde{L} = L + \lambda R$ rather than just the regularizer $R(\theta)$. In this setting there is an additional correction term to the Hessian (Proposition 1) that requires carefully controlling the value of the loss across reference points to prove convergence to a stationary point.

- The analysis in Blanc et al. [3] requires $\eta, \tau$ to be chosen in terms of the condition number of $\nabla^2 L$ which can quickly grow during training as $\nabla^2 L$ is changing. This makes it impossible to directly repeat the argument. We avoid this by precisely analyzing the error incurred by small eigenvalues, allowing us to prove convergence to an $(\epsilon, \gamma)$ stationary point of $\frac{1}{\lambda} \tilde{L}$ for fixed $\eta, \lambda$ even if the smallest nonzero eigenvalue of $\nabla^2 L$ converges to 0 during training.

- Unlike in Blanc et al. [3], we do not require the learning rate $\eta$ to be small. Instead, we only require that $\lambda$ scales with $\epsilon$ which can be accomplished either by decreasing the learning rate $\eta$ or increasing the batch size $B$. This allows for stronger implicit regularization in the setting when $\eta$ is large (see Section 6.1). In particular, our regularizer $R(\theta)$ changes with $\eta$ and is only equal to the regularizer in Blanc et al. [3] in the limit $\eta \to 0$.

## 3.2 Global Convergence

In order to prove convergence to an $(\epsilon, \gamma)$-stationary point of $\frac{1}{\eta}\nabla\tilde{L}$, we will define a sequence of reference points $\theta_m^*$ and coupling times $\{\tau_m\}$ and repeatedly use a version of Lemma 1 to describe the long term behavior of $\theta$. For notational simplicity, given a sequence of coupling times $\{\tau_m\}$, define $T_m = \sum_{k<m} \tau_k$ to be the total number of steps until we have reached the reference point $\theta_m^*$.

To be able to repeat the local analysis in Lemma 1 with multiple reference points, we need a more general coupling lemma that allows the random process $\xi$ defined in each coupling to continue where the random process in the previous coupling ended. To accomplish this, we define $\xi$ outside the scope of the local coupling lemma:

**Definition 3.** *Given a sequence of reference points $\{\theta_m^*\}$ and a sequence of coupling times $\{\tau_m\}$, we define the random process $\xi$ by $\xi_0 = 0$, and for $k \in [T_m, T_{m+1})$,*

$$\epsilon_k^* = \frac{\eta}{B} \sum_{i\in\mathcal{B}^{(k)}} \epsilon_i^{(k)} \nabla f_i(\theta_m^*) \qquad and \qquad \xi_{k+1} = (I - \eta G(\theta_m^*))\xi_k + \epsilon_k^*.$$

Then we can prove the following more general coupling lemma:

**Lemma 2.** *Let $\mathcal{X}, \mathcal{L}, \mathcal{D}, \mathcal{M}, \mathcal{T}$ be defined as in Lemma 1. Assume $f$ satisfies Assumption 1 and $\eta$ satisfies Assumption 2. Let $\Delta_m = \theta_{T_m} - \xi_{T_m} - \theta_m^*$ and assume that $\|\Delta_m\| \leq \mathcal{D}$ and $L(\theta_m^*) \leq \mathcal{L}$ for some $0 < \delta \leq 1/2$. Then for any $\tau_m \leq \mathcal{T}$ satisfying $\max_{k\in[T_m, T_{m+1})} \|\Phi_{k-T_m}(\theta_m^* + \Delta_m) - \theta_m^*\| \leq 8\mathcal{M}$, with probability at least $1 - 10d\tau_m e^{-\iota}$ we have simultaneously for all $k \in (T_m, T_{m+1}]$,*

$$\|\theta_k - \xi_k - \Phi_{k-T_m}(\theta_m^* + \Delta_m)\| \leq \mathcal{D}, \qquad \mathbb{E}[\xi_k] = 0, \qquad and \qquad \|\xi_k\| \leq \mathcal{X}.$$

Unlike in Lemma 1, we couple to the regularized trajectory starting at $\theta_m^* + \Delta_m$ rather than at $\theta_m^*$ to avoid accumulating errors (see Figure 2). The proof is otherwise identical to that of Lemma 1.

The proof of Theorem 1 easily follows from the following lemma which states that we decrease the regularized loss $\tilde{L}$ by at least $\mathcal{F}$ after every coupling:

**Lemma 3.** *Let $\mathcal{F} = \frac{\mathcal{D}^2}{\eta\nu\mathcal{T}}$. Let $\Delta_m = \theta_{T_m} - \xi_{T_m} - \theta_m^*$ and assume $\|\Delta_m\| \leq \mathcal{D}$ and $L(\theta_m^*) \leq \mathcal{L}$. Then if $\theta_{T_m}$ is not an $(\epsilon, \gamma)$-stationary point, there exists some $\tau_m < \mathcal{T}$ such that if we define*

$$\theta_{m+1}^* = \Phi_{\tau_n}(\theta_m^* + \Delta_m) \qquad and \qquad \Delta_{m+1} = \theta_{T_{m+1}} - \xi_{T_{m+1}} - \theta_{m+1}^*,$$

*then with probability $1 - 10d\tau_m e^{-\iota}$,*

$$\tilde{L}(\theta_{m+1}^*) \leq L(\theta_m^*) - \mathcal{F}, \qquad \|\Delta_{m+1}\| \leq \mathcal{D} \qquad and \qquad L(\theta_{m+1}^*) \leq \mathcal{L}.$$

We defer the proofs of Lemma 2 and Lemma 3 to Appendix B. Theorem 1 now follows directly from repeated applications of Lemma 3:

*Proof of Theorem 1.* By assumption there exists some $\theta_0^*$ such that $L(\theta_0^*) \leq \mathcal{L}$ and $\|\theta_0 - \theta_0^*\| \leq \mathcal{D}$. Then so long as $\theta_{T_m}$ is not an $(\epsilon, \gamma)$-stationary point, we can inductively apply Lemma 3 to get the existence of coupling times $\{\tau_m\}$ and reference points $\{\theta_m^*\}$ such that for any $m \geq 0$, with probability $1 - 10dT_m e^{-\iota}$ we have $\tilde{L}(\theta_m^*) \leq \tilde{L}(\theta_0^*) - m\mathcal{F}$. As $\tilde{L}(\theta_0^*) - \tilde{L}(\theta_m^*) = O(\lambda)$, this can happen for at most $m = O\left(\frac{\lambda}{\mathcal{F}}\right)$ reference points, so at most $T = O\left(\frac{\lambda\mathcal{T}}{\mathcal{F}}\right) = \tilde{O}\left(\eta^{-1}\lambda^{-1-\delta}\right)$ iterations of Algorithm 1. By the choice of $\iota$, this happens with probability $1 - 10dTe^{-\iota} \geq 1 - \zeta$. $\square$

## 4 Experiments

In order to test the ability of SGD with label noise to escape poor global minimizers and converge to better minimizers, we initialize Algorithm 1 at global minimizers of the training loss which achieve $100\%$ training accuracy yet generalize poorly to the test set. Minibatch SGD would remain fixed at these initializations because both the gradient and the noise in minibatch SGD vanish at any global minimizer of the training loss. We show that SGD with label noise escapes these poor initializations and converges to flatter minimizers that generalize well, which supports Theorem 1. We run experiments with two initializations:

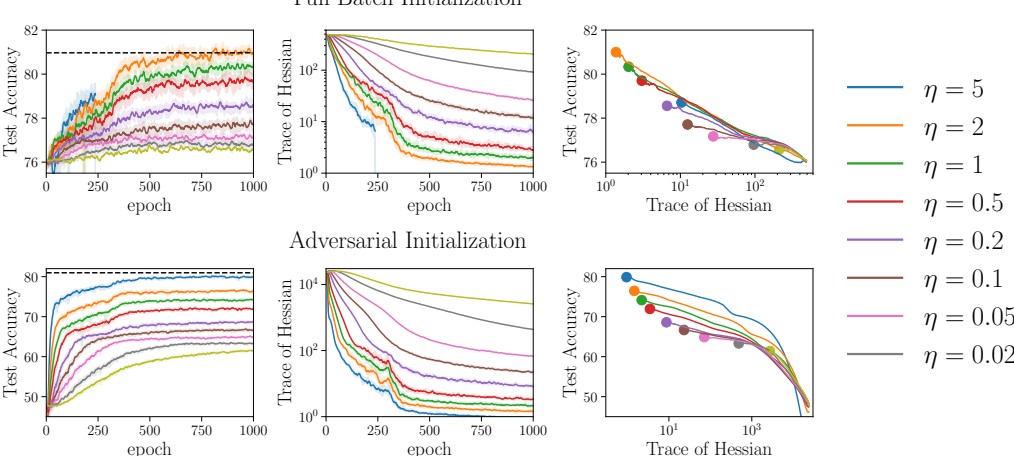

Figure 3: **Label Noise SGD escapes poor global minimizers.** The left column displays the training accuracy over time, the middle column displays the value of $\operatorname{tr} \nabla^2 L(\theta)$ over time which we use to approximate the implicit regularizer $R(\theta)$, and the right column displays their correlation. The horizontal dashed line represents the minibatch SGD baseline with random initialization. We report the median results over 3 random seeds and shaded error bars denote the min/max over the three runs. The correlation plot uses a running average of 100 epochs for visual clarity.

**Full Batch Initialization:** We run full batch gradient descent with random initialization until convergence to a global minimizer. We call this minimizer the full batch initialization. The final test accuracy of the full batch initialization was 76%.

**Adversarial Initialization:** Following Liu et al. [21], we generate an adversarial initialization with final test accuracy 48% that achieves zero training loss by first teaching the network to memorize random labels and then training it on the true labels. See Appendix D for full details.

Experiments were run with ResNet18 on CIFAR10 [17] without data augmentation or weight decay. The experiments were conducted with randomized label flipping with probability 0.2 (see Appendix E for the extension of Theorem 1 to classification with label flipping), cross entropy loss, and batch size 256. Because of the difficulty in computing the regularizer $R(\theta)$, we approximate it by its lower bound $\operatorname{tr} \nabla^2 L(\theta)$. Figure 3 shows the test accuracy and $\operatorname{tr} \nabla^2 L$ throughout training.

SGD with label noise escapes both zero training loss initializations and converges to flatter minimizers that generalize much better, reaching the SGD baseline from the fullbatch initialization and getting within 1% of the baseline from the adversarial initialization. The test accuracy in both cases is strongly correlated with $\operatorname{tr} \nabla^2 L$. The strength of the regularization is also strongly correlated with $\eta$, which supports Theorem 1.

## 5 Extensions

### 5.1 SGD with momentum

We replace the update in Algorithm 1 with heavy ball momentum with parameter $\beta$:

$$\theta_{k+1} = \theta_k - \eta \nabla \hat{L}^{(k)}(\theta_k) + \beta(\theta_k - \theta_{k-1}). \tag{5}$$

We define:

$$R(\theta) = \frac{1+\beta}{2\eta} \operatorname{tr} \log\left(1 - \frac{\eta}{2(1+\beta)}\nabla^2 L(\theta)\right), \qquad \lambda = \frac{\eta\sigma^2}{B(1-\beta)}, \tag{6}$$

and as before $\tilde{L}(\theta) = L(\theta) + \lambda R(\theta)$. Let

$$\Phi_0(\theta) = \theta, \qquad \Phi_{k+1}(\theta) = \Phi_k(\theta) - \eta \nabla \tilde{L}(\Phi_k(\theta)) + \beta(\Phi_k(\theta) - \Phi_{k-1}(\theta)) \tag{7}$$

represent gradient descent with momentum on $\tilde{L}$. Then we have the following local coupling lemma:

**Lemma 4.** *Let*

$$\mathcal{X} = \sqrt{\frac{2\lambda n^2 \iota}{\nu}}, \qquad \mathcal{L} = c\lambda^{1+\delta}, \qquad \mathcal{D} = c\sqrt{\mathcal{L}}\iota, \qquad \mathcal{T} = \frac{1}{c^2 \eta \mathcal{X} \iota}, \qquad (8)$$

*where $c$ is a sufficiently large constant. Assume $f$ satisfies Assumption 1 and $\eta \leq \frac{(2-\nu)(1+\beta)}{\ell}$. Let $\theta$ follow Algorithm 1 with momentum $\beta$ starting at $\theta^*$ and $L(\theta^*) \leq \mathcal{L}$ for some $0 < \delta \leq 1/2$. Then there exists a random process $\{\xi_k\}$ such that for any $\tau \leq \mathcal{T}$ satisfying $\max_{k \leq \tau} \|\Phi_k(\theta^*) - \theta^*\| \leq 8\mathcal{D}$, with probability at least $1 - 10d\tau e^{-\iota}$ we have simultaneously for all $k \leq \tau$,*

$$\|\theta_k - \xi_k - \Phi_k(\theta^*)\| \leq \mathcal{D}, \qquad \mathbb{E}[\xi_k] = 0, \qquad and \qquad \|\xi_k\| \leq \mathcal{X}. \qquad (9)$$

As in Lemma 1, the error is $8$ times smaller than the maximum movement of the regularized trajectory. Note that momentum increases the regularization parameter $\lambda$ by $\frac{1}{1-\beta}$. For the commonly used momentum parameter $\beta = 0.9$, this represents a $10\times$ increase in regularization, which is likely the cause of the improved performance in Figure 4 ($\beta = 0.9$) over Figure 3 ($\beta = 0$).

### 5.2 Arbitrary Noise Covariances

The analysis in Section 3.1 is not specific to label noise SGD and can be carried out for arbitrary noise schemes. Let $\theta$ follow $\theta_{k+1} = \theta_k - \eta \nabla L(\theta_k) + \epsilon_k$ starting at $\theta_0$ where $\epsilon_k \sim N(0, \eta\lambda\Sigma(\theta_k))$ and $\Sigma^{1/2}$ is Lipschitz. Given a matrix $S$ we define the regularizer $R_S(\theta) = \langle S, \nabla^2 L(\theta) \rangle$. The matrix $S$ controls the weight of each eigenvalue. As before we can define $\tilde{L}_S(\theta) = L(\theta) + \lambda R_S(\theta)$ and $\Phi^S_{k+1}(\theta) = \Phi^S_k(\theta) - \eta \nabla \tilde{L}_S(\Phi_k(\theta))$ to be the regularized loss and the regularized trajectory respectively. Then we have the following version of Lemma 1:

**Proposition 5.** *Let $\theta$ be initialized at a minimizer $\theta^*$ of $L$. Assume $\nabla^2 L$ is Lipschitz, let $H = \nabla^2 L(\theta^*)$ and assume that $\Sigma(\theta^*) \preceq CH$ for some absolute constant $C$. Let $\mathcal{X} = \sqrt{\frac{Cd\lambda\iota}{\nu}}$, $\mathcal{D} = c\lambda^{3/4}\iota$, and $\mathcal{T} = \frac{1}{c^2 \eta \mathcal{X} \iota}$ for a sufficiently large constant $c$. Then there exists a mean zero random process $\xi$ such that for any $\tau \leq \mathcal{T}$ satisfying $\max_{k < \tau} \|\Phi_k(\theta^*) - \theta^*\| \leq 8\mathcal{D}$ and with probability $1 - 10d\tau e^{-\iota}$, we have simultaneously for all $k \leq \tau$:*

$$\|\theta_k - \xi_k - \Phi^S_k(\theta_0)\| \leq \mathcal{D} \qquad and \qquad \|\xi_k\| \leq \mathcal{X},$$

*where $S$ is the unique fixed point of $S \leftarrow (I - \eta H)S(I - \eta H) + \eta\lambda\Sigma(\theta^*)$ restricted to $\mathrm{span}(H)$.*

As in Lemma 1, the error is $8$ times smaller than the maximum movement of the regularized trajectory. Although Proposition 5 couples to gradient descent on $R_S$, $S$ is defined in terms of the Hessian and the noise covariance at $\theta^*$ and therefore depends on the choice of reference point. Because $R_S$ is changing, we cannot repeat Proposition 5 as in Section 3.2 to prove convergence to a stationary point because there is no fixed potential. Although it is sometimes possible to relate $R_S$ to a fixed potential $R$, we show in Appendix F.2 that this is not generally possible by providing an example where minibatch SGD perpetually cycles. Exploring the properties of these continuously changing potentials and their connections to generalization is an interesting avenue for future work.

## 6 Discussion

### 6.1 Sharpness and the Effect of Large Learning Rates

Various factors can control the strength of the implicit regularization in Theorem 1. Most important is the implicit regularization parameter $\lambda = \frac{\eta\sigma^2}{|B|}$. This supports the hypothesis that large learning rates and small batch sizes are necessary for implicit regularization [9, 26], and agrees with the standard linear scaling rule which proposes that for constant regularization strength, the learning rate $\eta$ needs to be inversely proportional to the batch size $|B|$.

However, our analysis also uncovers an *additional* regularization effect of large learning rates. Unlike the regularizer in Blanc et al. [3], the implicit regularizer $R(\theta)$ defined in Equation (1) is dependent on $\eta$. It is not possible to directly analyze the behavior of $R(\theta)$ as $\eta \to 2/\lambda_1$ where $\lambda_1$ is the largest

eigenvalue of $\nabla^2 L$, as in this regime, $R(\theta) \to \infty$ (see Figure 1). If we let $\eta = \frac{2-\nu}{\lambda_1}$, then we can better understand the behavior of $R(\theta)$ by normalizing it by $\log 2/\nu$. This gives[2]

$$\frac{R(\theta)}{\log 2/\nu} = \sum_i \frac{R(\lambda_i)}{\log 2/\nu} = \|\nabla^2 L(\theta)\|_2 + O\left(\frac{1}{\log 2/\nu}\right) \xrightarrow{\nu \to 0} \|\nabla^2 L(\theta)\|_2$$

so after normalization, $R(\theta)$ becomes a better and better approximation of the spectral norm $\|\nabla^2 L(\theta)\|$ as $\eta \to 2/\lambda_1$. $R(\theta)$ can therefore be seen as interpolating between $\operatorname{tr} \nabla^2 L(\theta)$, when $\eta \approx 0$, and $\|\nabla^2 L(\theta)\|_2$ when $\eta \approx 2/\lambda_1$. This also suggests that SGD with large learning rates may be more resilient to the edge of stability phenomenon observed in Cohen et al. [4] as the implicit regularization works harder to control eigenvalues approaching $2/\eta$.

The sharpness-aware algorithm (SAM) of [7] is also closely related to $R(\theta)$. SAM proposes to minimize $\max_{\|\delta\|_2 \le \epsilon} L(\theta + \delta)$. At a global minimizer of the training loss,

$$\max_{\|\delta\|_2 \le \epsilon} L(\theta^* + \delta) = \max_{\|\delta\|_2 \le \epsilon} \frac{1}{2}\delta^\top \nabla^2 L(\theta^*)\delta + O(\epsilon^3) \approx \frac{\epsilon^2}{2}\|\nabla^2 L(\theta^*)\|_2.$$

The SAM algorithm is therefore explicitly regularizing the spectral norm of $\nabla^2 L(\theta)$, which is closely connected to the large learning rate regularization effect of $R(\theta)$ when $\eta \approx 2/\lambda_1$.

## 6.2 Generalization Bounds

The implicit regularizer $R(\theta)$ is intimately connected to data-dependent generalization bounds, which measure the Lipschitzness of the network via the network Jacobian. Specifically, Wei and Ma [30] propose the all-layer margin, which bounds the generalization error $\lesssim \frac{\sum_{l=1}^{L} \mathcal{C}_l}{\sqrt{n}}\sqrt{\frac{1}{n}\sum_{i=1}^{n}\frac{1}{m_F(x_i,y_i)^2}}$, where $\mathcal{C}_l$ depends only on the norm of the parameters and $m_F$ is the all-layer margin. The norm of the parameters is generally controlled by weight decay regularization, so we focus our discussion on the all-layer margin. Ignoring higher-order secondary terms, Wei and Ma [30, Heuristic derivation of Lemma 3.1] showed for a feed-forward network $f(\theta; x) = \theta_L \sigma(\theta_{L-1} \ldots \sigma(\theta_1 x))$, the all-layer margin satisfies[3]:

$$\frac{1}{m_F(x, y)} \lesssim \frac{\|\{\frac{\partial f}{\partial \theta_l}\}_{l \in [L]}\|_2}{\text{output margin of } (x, y)} \implies \text{generalization error} \lesssim \frac{\sum_{l=1}^{L} \mathcal{C}_l}{\sqrt{n}}\sqrt{\frac{R(\theta)}{\text{output margin}}}$$

as $R(\theta)$ is an upper bound on the squared norm of the Jacobian at any global minimizer $\theta$. We emphasize this bound is informal as we discarded the higher-order terms in controlling the all-layer margin, but it accurately reflects that the regularizer $R(\theta)$ lower bounds the all-layer margin $m_F$ up to higher-order terms. Therefore SGD with label noise implicitly regularizes the all-layer margin.

## Acknowledgments and Disclosure of Funding

AD acknowledges support from a NSF Graduate Research Fellowship. TM acknowledges support of Google Faculty Award and NSF IIS 2045685. JDL acknowledges support of the ARO under MURI Award W911NF-11-1-0303, the Sloan Research Fellowship, NSF CCF 2002272, and an ONR Young Investigator Award.

The experiments in this paper were performed on computational resources managed and supported by Princeton Research Computing, a consortium of groups including the Princeton Institute for Computational Science and Engineering (PICSciE) and the Office of Information Technology's High Performance Computing Center and Visualization Laboratory at Princeton University.

We would also like to thank Honglin Yuan and Jeff Z. HaoChen for useful discussions throughout various stages of the project.

---

[2]Here we assume $\lambda_1 > \lambda_2$. If instead $\lambda_1 = \ldots = \lambda_k > \lambda_{k+1}$, this limit will be $k\|\nabla^2 L(\theta)\|_2$.

[3]The output margin is defined as $\min_i f_i(\theta)y_i$. The following uses Equation (3.3) and the first-order approximation provided Wei and Ma [30] and the chain rule $\frac{\partial f}{\partial \theta_l} = \frac{\partial f}{\partial h_l}\frac{\partial h_l}{\partial \theta_{l-1}} = \frac{\partial f}{\partial h_l}h_{l-1}^\top$.

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
