## A  Limitations

In Section 2 we make three main assumptions: Assumption 1 (smoothness), Assumption 2 (learning rate separation), and Assumption 3 (KL).

Assumption 1 imposes the necessary smoothness conditions on $f$ to enable second order Taylor expansions of $\nabla L$. These smoothness conditions may not hold, e.g. if ReLU activations are used. This can be easily resolved by using a smooth activation like softplus or SiLU [13].

Assumption 2 is a very general assumption that lets $\eta$ be arbitrarily close to the maximum cutoff for gradient descent on a quadratic, $2/\ell$. However, for simplicity we do not track the dependence on $\nu$. This work therefore does not explain the ability of gradient descent to optimize neural networks at the "edge of stability" [4] when $\eta > 2/\ell$. Because we only assume Assumption 1 of the model, our results must apply to quadratics as a special case where any $\eta > 2/\ell$ leads to divergence so this assumption is strictly necessary.

Although Assumption 3 is very general (see Lemma 17), the specific value of $\delta$ plays a large role in our Theorem 1. In particular, if $L$ satisfies Assumption 3 for any $\delta \geq 1/2$ then the convergence rate in $\epsilon$ is $\epsilon^{-6}$. However, this convergence rate can become arbitrarily bad as $\delta \to 0$. This rate is driven by the bound on $E(\theta^*)$ in Proposition 1, which does not contribute to implicit regularization and cannot be easily controlled. The error introduced at every step from bounding $E(\theta)$ at a minimizer $\theta^*$ is $\tilde{O}(\eta\sqrt{\lambda L(\theta^*)})$ and the size of each step in the regularized trajectory is $\eta\lambda\|\nabla R(\theta^*)\|$. Therefore if $L(\theta^*) = \Omega(\lambda)$, the error term is greater than the movement of the regularized trajectory. Section 5.2 repeats the argument in Section 3.1 without making Assumption 3. However, the cost is that you can no longer couple to a fixed potential $R$ and instead must couple to a changing potential $R_S$.

One final limitation is our definition of stationarity (Definition 2). As we discuss in Section 2.3, this limitation is fundamental as the more direct statement of converging to an $\epsilon$-stationary point of $\frac{1}{\lambda}\tilde{L}$ is not true. Although we do not do so in this paper, if $\theta$ remains in a neighborhood of a fixed $\epsilon$-stationary point $\theta^*$ for a sufficiently long time, then it might be possible to remove this assumption by tail-averaging the iterates. However, this requires a much stronger notion of stationarity than first order stationarity which does not guarantee that $\theta$ remains in a neighborhood of $\theta^*$ for a sufficiently long time (e.g. it may converge to a saddle point which it then escapes).

## B  Missing Proofs

*Proof of Proposition 1.* We have

$$\nabla L(\theta) = \frac{1}{n}\sum_{i=1}^{n}(f_i(\theta) - y_i)\nabla f_i(\theta) \tag{10}$$

so

$$\nabla^2 L(\theta) = \frac{1}{n}\sum_{i=1}^{n}\left[\nabla f_i(\theta)\nabla f_i(\theta)^T + (f_i(\theta) - y_i)\nabla^2 f_i(\theta)\right] \tag{11}$$

$$= G(\theta) + E(\theta). \tag{12}$$

In addition if we define $e_i(\theta) = f_i(\theta) - y_i$,

$$\|E(\theta)\| = \frac{1}{n}\left\|\sum_{i=1}^{n}e_i(\theta)\nabla^2 f_i(\theta)\right\| \tag{13}$$

$$\leq \frac{1}{n}\left[\sum_{i=1}^{n}e_i(\theta)^2\right]^{1/2}\left[\sum_{i=1}^{n}\|\nabla^2 f_i(\theta)\|^2\right]^{1/2} \tag{14}$$

$$= \frac{1}{n}\sqrt{2nL(\theta)} \cdot \sqrt{n\rho_f^2} \tag{15}$$

$$= \sqrt{2\rho_f L(\theta)} \tag{16}$$

$$= O(\sqrt{L(\theta)}). \tag{17}$$

$\square$

**Definition 4.** *We define the quadratic variation $[\cdot]$ and quadratic covariation $[\cdot, \cdot]$ of a martingale $X$ to be*

$$[X]_k = \sum_{j<k} \|\xi_{j+1} - \xi_j\|^2 \qquad \text{and} \qquad [X, X]_k = \sum_{j<k} (\xi_{j+1} - \xi_j)(\xi_{j+1} - \xi_j)^T. \tag{18}$$

**Lemma 5** (Azuma-Hoeffding). *Let $X \in \mathbb{R}^d$ be a mean zero martingale with $[X]_k \leq \sigma^2$. Then with probability at least $1 - 2de^{-\iota}$,*

$$\|X_k\| \leq \sigma\sqrt{2\iota}. \tag{19}$$

**Corollary 1.** *Let $X \in \mathbb{R}^d$ be a mean zero martingale with $[X, X]_k \preceq M$. Then with probability at least $1 - 2de^{-\iota}$,*

$$\|X_k\| \leq \sqrt{2\operatorname{tr}(M)\iota}. \tag{20}$$

*Proof of Proposition 2.* A simple induction shows that

$$\xi_k = \sum_{j<k} (I - \eta G)^j \epsilon^*_{k-j-1}. \tag{21}$$

Then

$$\mathbb{E}[\xi_k \xi_k^T] = \sum_{j<k} (I - \eta G)^j \eta\lambda G (I - \eta G)^j \tag{22}$$

$$= \eta\lambda G(2\eta G - \eta^2 G^2)^\dagger (I - (I - \eta G)^{2k}) \tag{23}$$

$$= \lambda\Pi_G (2 - \eta G)^{-1}(I - (I - \eta G)^{2k}). \tag{24}$$

Therefore $\mathbb{E}[\xi_k \xi_k^T] \preceq \frac{\eta}{\nu} I$ and $\mathbb{E}[\xi_k \xi_k^T] \to \lambda\Pi_G(2 - \eta G)^{-1}$. The partial sums of Equation (21) form a martingale with quadratic covariation bounded by

$$\sum_{j<k} (I - \eta G)^j \epsilon^*_{k-j-1} (\epsilon^*_{k-j-1})^T (I - \eta G)^j \tag{25}$$

$$\preceq \sum_{j<k} (I - \eta G)^j n\eta\lambda G (I - \eta G)^j \tag{26}$$

$$= n\lambda\Pi_G (2 - \eta G)^{-1}(I - (I - \eta G)^{2k}) \tag{27}$$

$$\preceq \frac{n\lambda}{\nu} I \tag{28}$$

therefore by Corollary 1, with probability at least $1 - 2de^{-\iota}$, $\|\xi_k\| \leq \mathscr{X}$. $\qquad\square$

We prove the following version of Proposition 2 for the setting of Lemma 2:

**Proposition 6.** *Let $\xi_k$ be defined as in Definition 3. Then for any $t \geq 0$, with probability $1 - 2de^{-\iota}$, $\|\xi_t\| \leq \mathscr{X}$.*

*Proof.* For $k \in (T_m, T_{m+1}]$ define $G_k = G(\theta^*_m)$. Then we can write for any $k \geq 0$,

$$\xi_{k+1} = (I - \eta G_k)\xi_k + \epsilon^*_k. \tag{29}$$

Let $\mathcal{F}_t = \sigma\{\mathcal{B}^{(k)}, \epsilon^{(k)} : k < t\}$. To each $k$ we will associate a martingale $\{X_j^{(k)}\}_{j \leq k}$ adapted to $\mathcal{F}$ as follows. First let $X_0^{(k)} = 0$. Then for all $k \geq 0$ and all $j \geq 0$,

$$X_{j+1}^{(k)} = \begin{cases} (I - \eta G_{k-1})X_j^{(k-1)} & j < k-1 \\ X_j^{(k)} + \epsilon^*_{k-1} & j = k-1. \end{cases}$$

First we need to show $X^{(k)}$ is in fact a martingale. We will show this by induction on $k$. The base case of $k = 0$ is trivial. Next, it is easy to see that $X_j^{(k)} \in \mathcal{F}_j$. Therefore,

$$\mathbb{E}[X_k^{(k)}|\mathcal{F}_{k-1}] = \mathbb{E}[X_{k-1}^{(k)}|\mathcal{F}_{k-1}] = X_{k-1}^{(k)} \tag{30}$$

and for $j < k - 1$:

$$\mathbb{E}[X_{j+1}^{(k)}|\mathcal{F}_j] = (I - \eta G_{k-1})\mathbb{E}[X_{j+1}^{(k-1)}|\mathcal{F}_j] \tag{31}$$

$$= (I - \eta G_{k-1})X_j^{(k-1)} \tag{32}$$

$$= X_j^k \tag{33}$$

where the second line followed from the induction hypothesis and the third line followed from the definition of $X_j^{(k)}$. Therefore $X^{(k)}$ is a martingale for all $k$.

Next, I claim that $\xi_k = X_k^{(k)}$. We can prove this by induction on $k$. The base case is trivial as $\xi_0 = X_0^{(0)} = 0$. Then,

$$X_{k+1}^{(k+1)} = X_k^{(k+1)} + \epsilon_k^* \tag{34}$$

$$= (I - \eta G_k)X_k^{(k)} + \epsilon_k^* \tag{35}$$

$$= \xi_{k+1}. \tag{36}$$

Finally, I claim that $[X^{(k)}, X^{(k)}]_k \preceq \frac{n\lambda}{\nu}I$. We will prove this by induction on $k$. The base case is trivial as $X_0^{(0)} = 0$. Then,

$$[X^{(k+1)}, X^{(k+1)}]_{k+1} = [X^{(k+1)}, X^{(k+1)}]_k + \epsilon_k^*(\epsilon_k^*)^T \tag{37}$$

$$= (I - \eta G_k)[X^{(k)}, X^{(k)}]_k(I - \eta G_k) + \epsilon_k^*(\epsilon_k^*)^T \tag{38}$$

$$\preceq \frac{n\lambda}{\nu}\left[(I - \eta G_k)^2 + \eta\nu G_k\right] \tag{39}$$

$$\preceq \frac{n\lambda}{\nu}\left[I - G_k(2 - \eta G_k - \nu I)\right] \tag{40}$$

$$\preceq \frac{n\lambda}{\nu}I. \tag{41}$$

Therefore by Corollary 1, $\|\xi_k\| \leq \mathscr{X}$ with probability at least $1 - 2de^{-\iota}$. $\qquad\square$

We will prove Proposition 3 and Proposition 4 in the more general setting of Lemma 2. For notational simplicity we will apply the Markov property and assume that $m = 0$. We define $\Delta = \Delta_0$ and $\theta^* = \theta_0^*$ and note that due to this time change that $\xi_0$ is not necessarily 0. We define $v_k = \theta_k - \Phi_k(\theta^* + \Delta)$ and $r_k = \theta_k - \xi_k - \Phi_k(\theta^* + \Delta)$.

*Proof of Proposition 3.* First, by Proposition 6, $\|\xi_t\| \leq \mathscr{X}$ with probability at least $1 - 2de^{-\iota}$. Then note that for $k \leq t$,

$$\|\theta_k - \theta^*\| \leq \|\xi_k\| + \|r_k\| + \|\Phi_k(\theta^* + \Delta) - \theta^*\| = O(\mathscr{X}) \qquad \text{and} \qquad \theta_k - \theta^* = \xi_k + O(\mathscr{M}) \tag{42}$$

so Taylor expanding the update in Algorithm 1 and Equation (2) to second order around $\theta^*$ and subtracting gives

$$v_{k+1} = (I - \eta G)v_k + \epsilon_k^* + m_k + z_k \tag{43}$$

$$- \eta\left[\frac{1}{2}\nabla^3 L(\theta_k - \theta^*, \theta_k - \theta^*) - \frac{1}{2}\nabla^3 L(\Phi_k(\theta^*) - \theta^*, \Phi_k(\theta^*) - \theta^*) - \lambda\nabla R\right]$$

$$+ O(\eta\mathscr{X}(\sqrt{\mathscr{L}} + \mathscr{X}^2))$$

$$= (I - \eta G)v_k + \epsilon_k^* + m_k + z_k - \eta\left[\frac{1}{2}\nabla^3 L(\xi_k, \xi_k) - \lambda\nabla R\right] + O(\eta\mathscr{X}(\sqrt{\mathscr{L}} + \mathscr{M} + \mathscr{X}^2)).$$

Subtracting Equation (4), we have

$$r_{k+1} = (I - \eta G)r_k - \eta\left[\frac{1}{2}\nabla^3 L(\xi_k, \xi_k) - \lambda\nabla R\right] + m_k + z_k + O(\eta\mathscr{X}(\sqrt{\mathscr{L}} + \mathscr{M} + \mathscr{X}^2)) \tag{44}$$

$$= (I - \eta G)r_k - \eta\left[\frac{1}{2}\nabla^3 L(\xi_k, \xi_k) - \lambda\nabla R\right] + m_k + z_k + \tilde{O}(c^{5/2}\eta\lambda^{1+\delta/2}). \tag{45}$$

$\qquad\square$

*Proof of Proposition 4.* Note that for each $i \in \mathcal{B}^{(k)}$,

$$\|\epsilon_i^{(k)}(\nabla f_i(\theta) - \nabla f_i(\theta^*))\| \leq \sigma \rho_f \|\theta - \theta^*\|. \tag{46}$$

Therefore by Lemma 5, with probability $1 - 2de^{-\iota}$,

$$\left\| \sum_{j<k} (I - \eta G)^j z_{k-j} \right\| = O(\sqrt{\eta \lambda k \iota} \mathscr{X}). \tag{47}$$

Next, note that because $\|\nabla \ell_i(\theta)\| = O(L(\theta))$, by Lemma 5, with probability at least $1 - 2de^{-\iota}$,

$$\left\| \sum_{j<k} (I - \eta G)^j m_{k-j} \right\| = O(\sqrt{\eta \lambda k \iota} \sqrt{L(\theta)}). \tag{48}$$

Next, by a second order Taylor expansion around $\theta^*$ we have

$$\sqrt{L(\theta)} \leq O(\sqrt{\mathscr{L}} + \mathscr{X}) \tag{49}$$

so

$$r_{t+1} = -\eta \sum_{k \leq t} (I - \eta G)^{t-k} \left[ \frac{1}{2} \nabla^3 L(\xi_k, \xi_k) - \lambda \nabla R \right] \tag{50}$$

$$+ O\left( \sqrt{\eta \lambda t} \left( \sqrt{\mathscr{L}} + \mathscr{X} \right) + \eta t \mathscr{X} \left( \sqrt{\mathscr{L}} + \mathscr{M} + \mathscr{X}^2 \right) \right)$$

$$= -\eta \sum_{k \leq t} (I - \eta G)^{t-k} \left[ \frac{1}{2} \nabla^3 L(\xi_k, \xi_k) - \lambda \nabla R \right] + \tilde{O}\left( \frac{\lambda^{1/2 + \delta/2}}{\sqrt{c}} \right). \tag{51}$$

Now we will turn to concentrating $\xi_k \xi_k^T$. We will use the shorthand $g_i = \nabla f_i(\theta^*)$. Let

$$S^* = \lambda(2 - \eta \nabla^2 L)^{-1}, \qquad \bar{S} = \lambda(2 - \eta G)^{-1}, \qquad \text{and} \qquad S_k = \xi_k \xi_k^T. \tag{52}$$

It suffices to bound

$$\eta \sum_{k \leq t} (I - \eta G)^{t-k} \frac{1}{2} \nabla^3 L (S_k - S^*).$$

We can expand out $\nabla^3 L$ using the fact that $L$ is square loss to get

$$\frac{1}{2} \nabla^3 L (S_k - S^*) = \frac{1}{n} \sum_{i=1}^n \left( H_i(S_k - S^*) g_i + \frac{1}{2} g_i \operatorname{tr}\left[(S_k - S^*) H_i\right] \right) + O(\sqrt{\mathscr{L}} \mathscr{X}^2), \tag{53}$$

so it suffices to bound the contribution of the first two terms individually. Starting with the second term, we have $\operatorname{tr}\left[(S_k - S^*) H_i\right] = O(\mathscr{X}^2)$, so by Lemma 12,

$$\eta \frac{1}{n} \sum_{i=1}^n \sum_{k \leq t} (I - \eta G)^{t-k} g_i \operatorname{tr}\left[(S_k - S^*) H_i\right] = O\left( \sqrt{\eta t} \mathscr{X}^2 \right). \tag{54}$$

For the first term, note that

$$S^* - \bar{S} = \lambda \left[ (2 - \eta \nabla^2 L)^{-1} \left( (2 - \eta G) - (2 - \eta \nabla^2 L) \right) (2 - \eta G)^{-1} \right] = O(\eta \lambda \sqrt{\mathscr{L}}) \tag{55}$$

so this difference contributes at most $O(\eta^2 \lambda t \sqrt{\mathscr{L}}) = O(\eta t \mathscr{X} \sqrt{\mathscr{L}})$ so it suffices to bound

$$\frac{1}{n} \sum_{i=1}^n \sum_{k \leq t} (I - \eta G)^{t-k} H_i (S_k - \bar{S}) g_i. \tag{56}$$

Now note that

$$S_{k+1} = (I - \eta G) S_k (I - \eta G) + (I - \eta G) \xi_k (\epsilon_k^*)^T + \epsilon_k^* \xi_k (I - \eta G) + (\epsilon_k^*)(\epsilon_k^*)^T \tag{57}$$

and that[4]

$$\bar{S} = (I - \eta G)\bar{S}(I - \eta G) + \eta \lambda G. \tag{58}$$

Let $D_k = S_k - \bar{S}$. Then subtracting these two equations gives

$$D_{k+1} = (I - \eta G)D_k(I - \eta G) + (I - \eta G)\xi_k(\epsilon_k^*)^T + \epsilon_k^*\xi_k(I - \eta G) + ((\epsilon_k^*)(\epsilon_k^*)^T - \eta \lambda G).$$

Let $W_k = (I - \eta G)\xi_k(\epsilon_k^*)^T + \epsilon_k^*\xi_k^T(I - \eta G)$ and let $Z_k = ((\epsilon_k^*)(\epsilon_k^*)^T - \eta \lambda G)$ so that

$$D_{k+1} = (I - \eta G)D_k(I - \eta G) + W_k + Z_k.$$

Then,

$$D_k = (I - \eta G)^k D_0(I - \eta G)^k + \sum_{j<k}(I - \eta G)^{k-j-1}(W_j + Z_j)(I - \eta G)^{k-j-1}.$$

Substituting the first term gives

$$\eta \frac{1}{n}\sum_{i=1}^{n}\sum_{k\leq t}(I - \eta G)^{t-k}H_i(I - \eta G)^k D_0(I - \eta G)^k g_i = O(\sqrt{\eta t}\mathscr{X}^2) \tag{59}$$

so we are left with the martingale part in the second term. The final term to bound is therefore

$$\eta \frac{1}{n}\sum_{i=1}^{n}\sum_{k\leq t}(I - \eta G)^{t-k}H_i\left[\sum_{j<k}(I - \eta G)^{k-j-1}(W_j + Z_j)(I - \eta G)^{k-j-1}\right]g_i. \tag{60}$$

We can switch the order of summations to get

$$\eta \frac{1}{n}\sum_{i=1}^{n}\sum_{j\leq t}\sum_{k=j+1}^{t}(I - \eta G)^{t-k}H_i(I - \eta G)^{k-j-1}(W_j + Z_j)(I - \eta G)^{k-j-1}g_i. \tag{61}$$

Now if we extract the inner sum, note that

$$\sum_{k=j+1}^{t}(I - \eta G)^{t-k}H_i(I - \eta G)^{k-j-1}(W_j + Z_j)(I - \eta G)^{k-j-1}g_i \tag{62}$$

is a martingale difference sequence. Recall that

$$\epsilon_j^* = \frac{\eta}{B}\sum_{l\in\mathcal{B}^{(j)}}\epsilon_l^{(j)}g_l \tag{63}$$

First, isolating the $W$ term, we get

$$\sum_{k=j+1}^{t}(I - \eta G)^{t-k}H_i(I - \eta G)^{k-j}\xi_j(\epsilon_j^*)^T(I - \eta G)^{k-j-1}g_i \tag{64}$$

$$+ \sum_{k=j+1}^{t}(I - \eta G)^{t-k}H_i(I - \eta G)^{k-j-1}\epsilon_j^*\xi_j^T(I - \eta G)^{k-j}g_i.$$

$$= \frac{\eta}{B}\sum_{l\in\mathcal{B}^{(j)}}\epsilon_l^{(j)}\left[\sum_{k=j+1}^{t}(I - \eta G)^{t-k}H_i(I - \eta G)^{k-j}\xi_j g_l^T(I - \eta G)^{k-j-1}g_i \tag{65}\right.$$

$$\left. + \sum_{k=j+1}^{t}(I - \eta G)^{t-k}H_i(I - \eta G)^{k-j-1}g_l\xi_j^T(I - \eta G)^{k-j}g_i\right].$$

---

[4]This identity directly follows from multiplying both sides by $2 - \eta G$ and the fact that all of these matrices commute .

The inner sums are bounded by $O(\mathscr{X}\eta^{-1})$ by Lemma 14. Therefore by Lemma 5, with probability at least $1 - 2de^{-\iota}$, the contribution of the $W$ term in Equation (60) is at most $O(\sqrt{\eta\lambda k\iota}\mathscr{X}) = O(\sqrt{\eta k}\mathscr{X}^2)$. The final remaining term to bound is the $Z$ term in (60). We can write the inner sum as

$$\frac{\eta\lambda}{B^2}\sum_{k=j+1}^{t}(I-\eta G)^{t-k}H_i(I-\eta G)^{k-j-1}\left(\frac{1}{\sigma^2}\sum_{l_1,l_2\in\mathcal{B}^{(k)}}\epsilon_{l_1}^{(j)}\epsilon_{l_2}^{(j)}g_{l_1}g_{l_2}^T - G\right)(I-\eta G)^{k-j-1}g_i$$

$$\tag{66}$$

which by Lemma 14 is bounded by $O(\lambda)$. Therefore by Lemma 5, with probability at least $1 - 2de^{-\iota}$, the full contribution of $Z$ to Equation (60) is $O(\eta\lambda\sqrt{t\iota}) = O(\sqrt{\eta t}\mathscr{X}^2)$. Putting all of these bounds together we get with probability at least $1 - 10de^{-\iota}$,

$$\|r_{t+1}\| = O\left[\sqrt{\eta\mathscr{T}}\mathscr{X}(\sqrt{\mathscr{L}}+\mathscr{X})+\eta\mathscr{T}\mathscr{X}(\sqrt{\mathscr{L}}+\mathscr{M}+\mathscr{X}^2)\right]$$

$$= \tilde{O}\left(\frac{\lambda^{1/2+\delta/2}}{\sqrt{c}}\right).$$

$\square$

The following lemma is necessary for some of the proofs below:

**Lemma 6.** *Assume that $L(\theta) \leq \mathscr{L}$. Then for any $k \geq 0$, $L(\Phi_k(\theta)) \leq \mathscr{L}$.*

*Proof.* By induction it suffices to prove this for $k = 1$. Let $\theta' = \Phi_1(\theta)$. First consider the case when

$$\|\nabla L(\theta')\| \leq \left(\frac{\mathscr{L}}{\mu}\right)^{1/(1+\delta)}.\tag{67}$$

Then by Assumption 3, $L(\theta') \leq \mathscr{L}$ so we are done. Otherwise, note that

$$\|\nabla L(\theta)\| \geq \|\nabla L(\theta')\| - \ell\|\theta - \theta'\|\tag{68}$$
$$\geq \Omega(c\lambda) - \eta\ell\|\nabla L(\theta)\|\tag{69}$$

so $\|\nabla L(\theta)\| \geq \Omega(c\lambda)$ and therefore $\|\nabla\tilde{L}(\theta)\| \geq \Omega(c\lambda)$ Then by the standard descent lemma,

$$L(\theta') \leq L(\theta) - \eta\nabla\tilde{L}(\theta)^T\nabla L(\theta) + \frac{\eta^2\ell}{2}\|\nabla\tilde{L}(\theta)^2\|$$

$$\leq L(\theta) - \frac{\eta}{2}(2-\eta\ell)\|\nabla\tilde{L}(\theta)\|^2 + O(\eta\lambda\|\nabla L(\theta)\|)$$

$$= L(\theta) - \frac{\eta\nu}{2}\|\nabla\tilde{L}(\theta)\|^2 + O(\eta\lambda\|\nabla L(\theta)\|)$$

and for $c$ sufficiently large, the second term is larger than the third so $L(\theta') \leq L(\theta) \leq \mathscr{L}$. $\square$

We break the proof of Lemma 3 into a sequence of propositions. The idea behind Lemma 3 is to consider the trajectory $\Phi_k(\theta_m^*)$, for $k \leq \mathscr{T}$. First, we want to carefully pick $\tau_m$ so that $\eta\sum_{k<\tau_m}\|\nabla\tilde{L}(\Phi_k(\theta_m^*))\|$ is sufficiently large to decrease the regularized loss $\tilde{L}$ but sufficiently small to be able to apply Lemma 2:

**Proposition 7.** *In the context of Lemma 3, if $\theta_{T_m}$ is not an $(\epsilon, \gamma)$-stationary point, there exists $\tau_m \leq \mathscr{T}$ such that:*

$$5\mathscr{M} \geq \eta\sum_{k<\tau_n}\|\nabla\tilde{L}(\Phi_k(\theta_n^*))\| \geq 4\mathscr{M}.\tag{70}$$

We can use this to lower bound the decrease in $\tilde{L}$ from $\theta_m^*$ to $\Phi_{\tau_m}(\theta_m^*)$:

**Proposition 8.** $\tilde{L}(\Phi_{\tau_m}(\theta_m^*)) \leq \tilde{L}(\theta_m^*) - 8\frac{\mathscr{D}^2}{\eta\nu\tau_m}$.

We now bound the increase in $\tilde{L}$ from $\Phi_{\tau_m}(\theta_m^*)$ to $\theta_{m+1}^*$. This requires relating the regularized trajectories starting at $\theta_m^*$ and $\theta_m^* + \Delta_m$. The following proposition shows that the two trajectories converge in the directions where the eigenvalues of $G(\theta_m^*)$ are large:

**Proposition 9.** *Let $G = G(\theta_m^*)$ and let $\tau_m$ be chosen as in Proposition 7. Then, $\theta_{m+1}^* - \Phi_{\tau_m}(\theta_m^*) = (I - \eta G)^{\tau_m}\Delta_m + r$ where $\|r\| = O(\eta\tau_m\mathscr{M}^2)$ and $\|r\|_G^2 = O(\eta\tau_m\mathscr{M}^4)$.*

Substituting the result in Proposition 9 into the second order Taylor expansion of $\tilde{L}$ centered at $\Phi_{\tau_m}(\theta_m^*)$ gives:

**Proposition 10.** $\tilde{L}(\theta_{m+1}^*) \leq \tilde{L}(\Phi_{\tau_m}(\theta_m^*)) + 7\frac{\mathscr{D}^2}{\eta\nu\tau_m}$

Combining Propositions 8 and 10, we have that

$$\tilde{L}(\theta_{m+1}^*) - \tilde{L}(\theta_m^*) \leq -\frac{\mathscr{D}^2}{\eta\nu\tau_m} \leq -\mathscr{F}. \tag{71}$$

where the last line follows from $\tau_m \leq \mathscr{T}$ and the definition of $\mathscr{F}$. Finally, the following proposition uses this bound on $\tilde{L}$ and Assumption 3 to bound $L(\theta_{m+1}^*)$:

**Proposition 11.** $L(\theta_{m+1}^*) \leq \mathscr{L}.$

The following corollary also follows from the choice of $\tau_m$, Proposition 9, and Lemma 2:

**Corollary 2.** $\|\Phi_{\tau_m}(\theta_m^* + \Delta_m) - \theta_m^*\| \leq 8\mathscr{M}$ *and with probability at least $1 - 8d\tau_m e^{-\iota}$, $\|\Delta_{m+1}\| \leq \mathscr{D}$.*

The proof of Lemma 3 follows directly from Equation (71), Proposition 11, and Corollary 2. The proofs of the above propositions can be found below:

*Proof of Proposition 7.* First, assume that

$$\eta \sum_{k < \mathscr{T}} \|\nabla\tilde{L}(\Phi_k(\theta_m^*))\| \geq 4\mathscr{M}. \tag{72}$$

Then we can upper bound each element in this sum by

$$\eta\|\nabla\tilde{L}(\Phi_k(\theta_m^*))\| \leq \eta\|\nabla L(\Phi_k(\theta_m^*))\| + \eta\lambda\|\nabla R(\Phi_k(\theta_m^*))\|. \tag{73}$$

Note that

$$\|\nabla L(\theta)\| = \left\|\frac{1}{n}\sum_{i=1}^n (f_i(\theta) - y_i)\nabla f_i(\theta)\right\| \tag{74}$$

$$\leq \frac{1}{n}\left[\sum_{i=1}^n (f_i(\theta) - y_i)^2\right]^{1/2}\left[\sum_{i=1}^n \|\nabla f_i(\theta)\|^2\right]^{1/2} \tag{75}$$

$$\leq \sqrt{2\ell_f L(\theta)} \tag{76}$$

and because $\nabla R$ is bounded,

$$\eta\|\nabla\tilde{L}(\Phi_k(\theta_m^*))\| \leq O(\eta L(\Phi_k(\theta_m^*)) + \eta\lambda). \tag{77}$$

Then by Lemma 6,

$$\eta\|\nabla\tilde{L}(\Phi_k(\theta_m^*))\| \leq O(\eta\sqrt{\mathscr{L}} + \eta\lambda) \leq \mathscr{M} \tag{78}$$

for sufficiently large $c$. Therefore there must exist $\tau_m$ such that

$$5\mathscr{M} \geq \eta \sum_{k < \mathscr{T}} \|\nabla\tilde{L}(\Phi_k(\theta_m^*))\| \geq 4\mathscr{M}. \tag{79}$$

Otherwise,

$$\eta \sum_{k < \mathscr{T}} \|\nabla\tilde{L}(\Phi_k(\theta_m^*))\| < 4\mathscr{M}. \tag{80}$$

Therefore there must exist some $k$ such that

$$\frac{1}{\lambda}\|\nabla\tilde{L}(\Phi_k(\theta_m^*))\| < \frac{4\mathscr{M}}{\eta\lambda\mathscr{T}} = O(\lambda^{\delta/2}\iota^{3/2}) \leq \epsilon \tag{81}$$

by the choice of $\lambda$ in Theorem 1. In addition,

$$\|\theta_{T_m} - \Phi_k(\theta_m^*)\| \leq \|\theta_{T_m} - \theta_m^*\| + 4\mathscr{M} \leq \mathscr{X} + \mathscr{D} + 4\mathscr{M} \leq \gamma \tag{82}$$

again by the choice of $\lambda$. Therefore $\theta_{T_m}$ is an $(\epsilon, \gamma)$-stationary point. $\qquad\square$

*Proof of Proposition 8.* We have by the standard descent lemma

$$\tilde{L}(\Phi_{\tau_m}(\theta_m^*)) \leq -\frac{\eta\nu}{2}\sum_{k<\tau_m}\|\nabla\tilde{L}(\Phi_{\tau_k}(\theta_m^*))\|^2 \tag{83}$$

$$\leq -\frac{\eta\nu}{2\tau_m}\left[\sum_{k<\tau_m}\|\nabla\tilde{L}(\Phi_{\tau_k}(\theta_m^*))\|\right]^2 \tag{84}$$

$$\leq -\frac{\nu\mathscr{M}^2}{2\eta\tau_m} \tag{85}$$

$$= -8\frac{\mathscr{D}^2}{\eta\nu\tau_m}. \tag{86}$$

$\square$

*Proof of Proposition 9.* Let $v_k = \Phi_k(\theta_m^* + \Delta_m) - \Phi_k(\theta_m^*)$, so that $v_0 = \Delta_m$ and let $r_k = v_k - (I - \eta G)^{\tau_m}$ so that $r_0 = 0$. Let $C$ be a sufficiently large absolute constant. We will prove by induction that $r_k \leq C\eta\tau_m\mathscr{M}^2$. Note that

$$\|\Phi_k(\theta_m^* + \Delta_m) - \theta_m^*\| \leq \|\Delta_m\| + \|\Phi_k(\theta_m^*) - \theta_m^*\| + \|v_k\| \tag{87}$$

$$\leq O(\mathscr{M} + \eta\mathscr{T}\mathscr{M}^2) \tag{88}$$

$$= O(\mathscr{M}) \tag{89}$$

because of the values chosen for $\mathscr{M}$, $\mathscr{T}$. Therefore Taylor expanding around $\theta_m^*$ gives:

$$v_{k+1} = v_k - \eta\left[\nabla\tilde{L}(\Phi_k(\theta_m^* + \Delta_m)) - \nabla\tilde{L}(\Phi_k(\theta_m^*))\right] \tag{90}$$

$$= v_k - \eta\nabla^2\tilde{L}v_k + O(\eta\mathscr{M}^2) \tag{91}$$

$$= (I - \eta G)v_k + O(\eta\mathscr{M}^2 + \eta\lambda\mathscr{M} + \eta\sqrt{\mathscr{L}}\mathscr{M}) \tag{92}$$

$$= (I - \eta G)v_k + s_k \tag{93}$$

where $\|s_k\| = O(\eta\mathscr{M}^2)$ by the definition of $\mathscr{M}$. Therefore

$$v_k = (I - \eta G)^k\Delta_m + O(\eta k\mathscr{M}^2). \tag{94}$$

In addition,

$$r_k = \sum_{j<k}(I - \eta G)^j s_{k-j} \tag{95}$$

so if $g_i = \nabla f_i(\theta_m^*)$,

$$r_k^T G r_k = \frac{1}{n}\sum_{i=1}^{n}(s_{k-j}^T\sum_{j<k}(I - \eta G)^j g_i)^2 \tag{96}$$

$$\leq O(\eta^2\mathscr{M}^4)\frac{1}{n}\sum_{i=1}^{n}\|\sum_{j<k}(I - \eta G)^j g_i\|^2 \tag{97}$$

$$= O(\eta\tau_m\mathscr{M}^4) \tag{98}$$

by Lemma 12, so we are done. $\qquad\square$

We will need the following lemma before the next proof:

**Lemma 7.** *For any $k < \tau_m$,*

$$\|\nabla \tilde{L}(\Phi_k(\theta_m^*))\| \geq 11\|\nabla \tilde{L}(\Phi_{\tau_m}(\theta_m^*))\|/12. \tag{99}$$

*Proof.*

$$\nabla \tilde{L}(\Phi_{k+1}(\theta)) = (I - \eta \nabla^2 L(\Phi_k(\theta)))\nabla \tilde{L}(\Phi_k(\theta)) + O(\eta^2\|\nabla \tilde{L}(\Phi_k(\theta))\|^2) \tag{100}$$

By Lemma 6 and Proposition 1,

$$\|I - \eta \nabla^2 L(\Phi_k(\theta))\| \leq 1 + \eta\sqrt{2\rho_f \mathscr{L}}.$$

In addition,

$$\|\nabla \tilde{L}(\Phi_k(\theta))\| = O(\lambda + \sqrt{\mathscr{L}}) \tag{101}$$

so

$$\|\nabla \tilde{L}(\Phi_{k+1}(\theta))\| \leq (1 + O(\mathscr{L}))\|\nabla \tilde{L}(\Phi_k(\theta))\|. \tag{102}$$

Therefore,

$$\|\nabla \tilde{L}(\Phi_{\tau_m}(\theta))\| \leq (1 + O(\mathscr{L}))^{\tau_m - k}\|\nabla \tilde{L}(\Phi_k(\theta))\| \tag{103}$$

$$\leq \exp(O(\mathscr{T}\mathscr{L}))\|\nabla \tilde{L}(\Phi_k(\theta))\| \tag{104}$$

$$\leq 12\|\nabla \tilde{L}(\Phi_k(\theta))\|/11 \tag{105}$$

for sufficiently large $c$. $\qquad\square$

*Proof of Proposition 10.* Let $v = \theta_{m+1} - \Phi_{\tau_m}(\theta_m^*) = (I - \eta G)^{\tau_m}\Delta_m + r$ where by Proposition 9, $\|r\| = O(\eta\tau_m\mathscr{M}^2)$, $G = G(\theta_m^*)$, and $r^T G r = O(\eta\tau_m\mathscr{M}^4)$. Then,

$$\tilde{L}(\theta_{m+1}^*) - \tilde{L}(\Phi_{\tau_m}(\theta_m^*)) \tag{106}$$

$$\leq \|v\|\|\tilde{L}(\Phi_{\tau_m}(\theta_m^*))\| + \frac{1}{2}v^T\nabla^2\tilde{L}(\Phi_{\tau_m}(\theta_m^*))v + O(\|v\|^3) \tag{107}$$

$$\leq \|v\|\|\tilde{L}(\Phi_{\tau_m}(\theta_m^*))\| + \frac{1}{2}v^T G v + O(\mathscr{D}^2(\mathscr{D} + \sqrt{\mathscr{L}} + \lambda)) \tag{108}$$

$$\leq \|v\|\|\tilde{L}(\Phi_{\tau_m}(\theta_m^*))\| + \Delta_m^T(I - \eta G)^{\tau_m}G(I - \eta G)^{\tau_m}\Delta_m + r^T G r \tag{109}$$

$$+ O(\mathscr{D}^2(\mathscr{D} + \sqrt{\mathscr{L}} + \lambda)). \tag{110}$$

By Proposition 9,

$$\|v\| \leq \mathscr{D} + O(\eta\tau_m\mathscr{M}^2) = \mathscr{D} + \mathscr{D} \cdot O\left(\frac{\lambda^{\delta/2}}{\sqrt{c}}\right) \leq 11\mathscr{D}/10$$

for sufficiently large $c$. Therefore by Lemma 7 and Proposition 7,

$$\|v\|\|\tilde{L}(\Phi_{\tau_m}(\theta_m^*))\| \leq \mathscr{D}\frac{6}{5\tau_m}\sum_{k<\tau_m}\|\tilde{L}(\Phi_k(\theta_m^*))\| \leq \frac{6\mathscr{D}^2}{\eta\nu\tau_m}. \tag{111}$$

By Lemma 10,

$$\Delta_m^T(I - \eta G)^{\tau_m}G(I - \eta G)^{\tau_m}\Delta_m \leq \frac{\mathscr{D}^2}{2\eta\nu\tau_m}. \tag{112}$$

By Proposition 9,

$$r^T G r = O(\eta\tau_m\mathscr{M}^4) = \frac{\mathscr{D}^2}{\eta\nu\tau_m}O(\eta^2\tau_m^2\mathscr{D}^2) \tag{113}$$

$$= \frac{\mathscr{D}^2}{\eta\nu\tau_m}O\left(\frac{\lambda^\delta}{c}\right) \tag{114}$$

$$\leq \frac{\mathscr{D}^2}{4\eta\nu\tau_m} \tag{115}$$

for sufficiently large $c$. Finally, the remainder term is bounded by

$$\frac{\mathscr{D}^2}{\eta\nu\tau_m} \cdot O(\eta\tau_m\mathscr{D}) \leq \frac{\mathscr{D}^2}{4\eta\nu\tau_m} \tag{116}$$

for sufficiently large $c$ for the same reason as above. Putting it all together,

$$\tilde{L}(\theta^*_{m+1}) - \tilde{L}(\Phi_{\tau_m}(\theta^*_m)) \leq \frac{6\mathscr{D}^2}{\eta\nu\tau_m} + \frac{\mathscr{D}^2}{2\eta\nu\tau_m} + \frac{\mathscr{D}^2}{4\eta\nu\tau_m} + \frac{\mathscr{D}^2}{4\eta\nu\tau_m} = \frac{7\mathscr{D}^2}{\eta\nu\tau_m}. \tag{117}$$

$\square$

*Proof of Proposition 11.* Assume otherwise for the sake of contradiction. Because $\nabla R$ is Lipschitz, $R(\theta^*_{m+1}) - R(\theta^*_m) = O(\mathscr{M})$. Therefore by Equation (71),

$$L(\theta^*_{m+1}) \leq L(\theta^*_m) - \frac{\mathscr{D}^2}{\eta\nu\tau_m} + O(\lambda\mathscr{M}).$$

Therefore we must have $\mathscr{D} = O(\eta\lambda\tau_m)$ so by Proposition 7 and Lemma 7 we have that $\|\nabla\tilde{L}(\Phi_{\tau_m}(\theta^*_m))\| = O(\lambda)$ and because $\lambda\nabla R = O(\lambda)$ we must have $\|\nabla L(\Phi_{\tau_m}(\theta^*_m))\| = O(\lambda)$. Therefore by Assumption 3,

$$L(\Phi_{\tau_m}(\theta^*_m)) = O(\lambda^{1+\delta}). \tag{118}$$

Then by the same arguments as in Proposition 10, we can Taylor expand around $\Phi_{\tau_m}(\theta^*_m)$ to get

$$L(\theta^*_{m+1}) - L(\Phi_{\tau_m}(\theta^*_m)) \tag{119}$$

$$\leq \|\nabla L(\Phi_{\tau_m}(\theta^*_m))\|v + \frac{1}{2}v^T\nabla^2 L(\Phi_{\tau_m}(\theta^*_m))v + O(\mathscr{D}^3) \tag{120}$$

$$\leq O\left(\lambda\mathscr{D} + \frac{\mathscr{D}^2}{\eta\tau} + \mathscr{D}^3\right) \tag{121}$$

$$\leq O(\lambda^{1+\delta}) \tag{122}$$

because $\delta \leq 1/2$. Therefore $L(\theta^*_{m+1}) = O(\lambda^{1+\delta}) \leq \mathscr{L}$ for sufficiently large $c$. $\square$

## C  Reaching a global minimizer with NTK

It is well known that overparameterized neural networks in the kernel regime trained by gradient descent reach global minimizers of the training loss [14, 5]. In this section we describe how to extend the proof in [5] to show that SGD with label noise (Algorithm 1) converges to a neighborhood of a global minimizer $\theta^*$ as required by Theorem 1. We will use the following lemma from [5]:

**Lemma 8** ([5], Lemma B.4). *There exists $R = \tilde{O}(\sqrt{m}\lambda_0)$ such that every $\theta \in B_R(\theta_0)$ satisfies $\lambda_{min}(\mathcal{G}(\theta)) \geq \lambda_0/2$ where $\mathcal{G}_{ij}(\theta) = \langle\nabla f_i(\theta), \nabla f_j(\theta)\rangle$ and $\lambda_0$ is the minimum eigenvalue of the infinite width NTK matrix.*

Let $\xi_0 = 0$ and $\theta^*_0 = \theta_0$. We will define $\xi_k, \theta^*_k$ iteratively as follows:

$$\xi_{k+1} = (I - \eta G(\theta^*_k))\xi_k + \epsilon_k \qquad \text{and} \qquad \theta^*_{k+1} = \theta^*_k - \eta\nabla L(\theta^*_k) - \eta E(\theta^*_k)(\theta_k - \theta^*_k) - z_k. \tag{123}$$

Let $v_k = \theta_k - \theta^*_k$ and let $r_k = v_k - \xi_k$. We will prove by induction that for all $t \leq T = \frac{4\log[L(\theta_0)\lambda_0/\lambda^2]}{\eta\lambda_0}$ we have $\|r_k\| \leq \mathscr{D}$. The base case follows from $r_0 = 0$. For $k \geq 0$ we have

$$v_{k+1} = v_k - \eta[\nabla L(\theta_k) - \nabla L(\theta^*_k) - E(\theta^*_k)v_k] + \epsilon^*_k$$
$$= (I - \eta G)v_k + \epsilon^*_k + O(\eta\mathscr{X}^2)$$

so

$$r_{k+1} = (I - \eta G)r_k + O(\eta\mathscr{X}^2)$$
$$= O(\eta T\mathscr{X}^2)$$
$$= O(\mathscr{D})$$

which completes the induction. Therefore it suffices to show that the loss of $\theta_T^*$ is small. We have

$$
\begin{aligned}
\mathbb{E}[L(\theta_{k+1}^*)|\theta_k^*] &\leq L(\theta_k^*) - \nabla L(\theta_k^*)^T[\eta\nabla L + \eta E(\theta_k^*)v_k] \\
&\quad + O[\eta^2\|\nabla L\|^2 + \eta^2\|E(\theta_k^*)\|^2\|v_k\|^2 + \|\epsilon_k - \epsilon_k^*\|^2] \\
&\leq L(\theta_k^*) - \frac{\eta}{4}\|\nabla L(\theta_k^*)\|^2 + O\left(\eta\|E(\theta_k^*)\|^2\|v_k\|^2 + \|\epsilon_k - \epsilon_k^*\|^2\right)
\end{aligned}
$$

where the last line follows from Young's inequality. Therefore,

$$
\begin{aligned}
L(\theta_{k+1}^*) &\leq L(\theta_k^*) - \frac{\eta}{4}\|\nabla L(\theta_k^*)\|^2 + O\left(\eta L(\theta_k^*)\mathscr{X}^2 + \eta\lambda\mathscr{X}^2\right) \\
&= L(\theta_k^*) - \frac{\eta}{4}\|\nabla L(\theta_k^*)\|^2 + \tilde{O}\left(\eta\lambda L(\theta_k^*) + \eta\lambda^2\right) \\
&= (1 + \tilde{O}(\eta\lambda))L(\theta_k^*) - \frac{\eta}{4}\|\nabla L(\theta_k^*)\|^2 + \tilde{O}\left(\eta\lambda^2\right).
\end{aligned}
$$

Let $J$ be the Jacobian of $f$ and $e$ be the vector of residuals. Then $\nabla L = Je$. Now so long as $\|\theta_k^* - \theta_0\| \leq R$,

$$
\|\nabla L(\theta_k^*)\|^2 = e(\theta_k^*)^T J(\theta_k^*)^T J(\theta_k^*)e(\theta_k^*) \geq \lambda_0\|e(\theta_k^*)\|^2 = 2\lambda_0 L(\theta_k^*). \tag{124}
$$

Therefore,

$$
L(\theta_{k+1}^*) \leq \left(1 - \frac{\eta\lambda_0}{2} + \tilde{O}(\eta\lambda)\right)L(\theta_k^*) + \tilde{O}\left(\eta\lambda^2\right).
$$

Now for $\lambda = \tilde{O}(\lambda_0)$,

$$
L(\theta_{k+1}^*) \leq \left(1 - \frac{\eta\lambda_0}{4}\right)L(\theta_k^*) + \tilde{O}\left(\eta\lambda^2\right)
$$

so

$$
\begin{aligned}
L(\theta_T^*) &\leq \left(1 - \frac{\eta\lambda_0}{4}\right)^T L(\theta_0) + \tilde{O}\left(\frac{\lambda^2}{\lambda_0}\right) \\
&\leq \tilde{O}\left(\frac{\lambda^2}{\lambda_0}\right) = O(\lambda^{1+\delta})
\end{aligned}
$$

for small $\lambda$ by the choice of $T$. It only remains to check that $\|\theta_k^* - \theta_0\| \leq R$. Note that

$$
\|\theta_k^* - \theta_0\| \leq \eta\sum_{j<k}\|\nabla L(\theta_j^*)\| + \tilde{O}(\eta T\sqrt{\lambda} + \sqrt{\eta T\lambda^2}) \tag{125}
$$

$$
\leq \tilde{O}\left(\eta\sum_{j\leq k}\sqrt{L(\theta_j^*)} + \sqrt{\lambda}\right) \tag{126}
$$

$$
\leq \tilde{O}\left(\frac{L(\theta_0)}{\lambda_0}\right) \tag{127}
$$

so for $m \geq \tilde{\Omega}(1/\lambda_0^4)$ we are done.

Note that a direct application of Theorem 1 requires starting $\xi$ at 0. However, this does not affect the proof in any way and the $\xi$ from this proof can simply be continued.

Finally, note that although $\|\frac{1}{\lambda}\nabla\tilde{L}(\theta)\| = O(1/\sqrt{m})$ at any global minimizer, Theorem 1 guarantees that for any $\lambda > 0$ we can find a point $\theta$ where $\|\frac{1}{\lambda}\nabla\tilde{L}(\theta)\| \lesssim \lambda^{\delta/2} \ll 1/\sqrt{m}$, as $m$ only needs to be larger than a fixed constant depending on the condition number of the infinite width NTK kernel.

## D   Additional Experimental Details

The model used in our experiments is ResNet18 with GroupNorm instead of BatchNorm to maintain independence of sample gradients when computed in a batch. We used a fixed group size of 32.

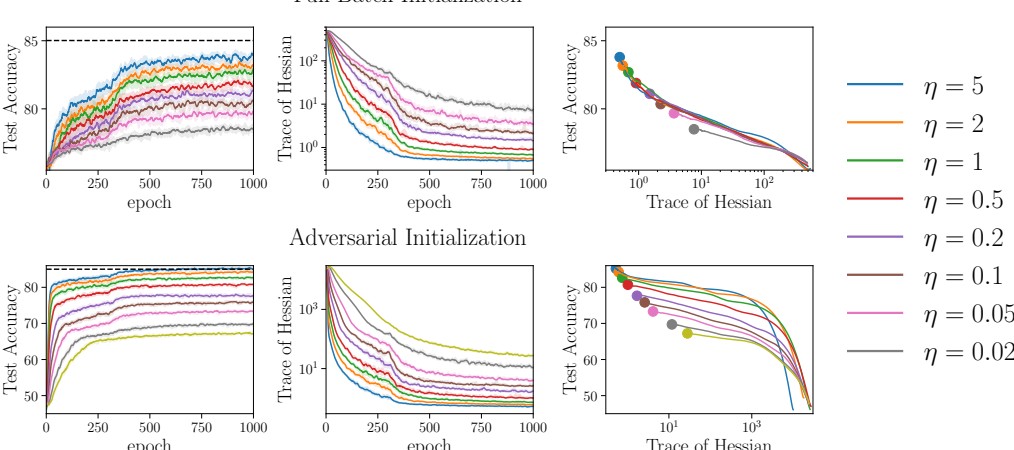

Figure 4: **Label Noise SGD with Momentum** ($\beta = 0.9$) The left column displays the training accuracy over time, the middle column displays the value of $\operatorname{tr} \nabla^2 L(\theta)$ over time which we use to approximate the implicit regularizer $R(\theta)$, and the right column displays their correlation. The horizontal dashed line represents the minibatch SGD baseline with random initialization. We report the median results over 3 random seeds and shaded error bars denote the min/max over the three runs. The correlation plot uses a running average of 100 epochs for visual clarity.

For the full batch initialization, we trained ResNet18 on the CIFAR10 training set (50k images, 5k per class) [17], with cross entropy loss. CIFAR10 images are provided under an MIT license. We trained using SGD with momentum with $\eta = 1$ and $\beta = 0.9$ for 2000 epochs. We used learning rate warmup starting at 0 which linearly increased until $\eta = 1$ at epoch 600 and then it decayed using a cosine learning rate schedule to 0 between epochs 600 and 2000. We also used a label smoothing value of 0.2 (non-randomized) so that the expected objective function is the same for when we switch to SGD with label flipping (see Appendix E). The final test accuracy was 76%.

For the adversarial initialization, we first created an augmented adversarial dataset as follows. We duplicate every image in CIFAR10 $10\times$, for a total of 500k images. In each image, we randomly zero out 10% of the pixels in the image and we assign each of the 500k images a random label. We trained ResNet18 to interpolate this dataset without label smoothing with the following hyperparameters: $\eta = 0.01$, 300 epochs, batch size 256. Starting from this initialization we ran SGD on the true dataset with $\eta = 0.01$ and a label smoothing value of 0.2 with batch size 256 for 1000 epochs. The final test accuracy was 48%.

For the remaining experiments starting at these two initializations we ran both with and without momentum (see Figure 4 for the results with momentum) for 1000 epochs per run. We used a fixed batch size of 256 and varied the maximum learning rate $\eta$. We used learning rate warmup by linearly increasing the learning rate from 0 to the max learning rate over 300 epochs, and we kept the learning rate constant from epochs 300 to 1000. The regularizer was estimated by computing the strength of the noise in each step and then averaging over an epoch. More specifically, we compute the average of $\|\nabla \hat{L}^{(k)}(\theta_k) - \nabla L^{(k)}(\theta_k)\|^2$ over an epoch and then renormalize by the batch size.

The experiments were run on a university cluster using NVIDIA P100 GPUs. Code was written in Python using PyTorch [24] and PyTorch Lightning [6], and experiments were logged using Wandb [2]. Code can be found at `https://github.com/adamian98/LabelNoiseFlatMinimizers`.

# E    Extension to Classification

We restrict $y_i \in \{-1, 1\}$, let $l : \mathbb{R} \to \mathbb{R}^+$ be an arbitrary loss function, and $p \in (0, 1)$ be a smoothing factor. Examples of $l$ include logistic loss, exponential loss, and square loss (see Table 1). We define $\bar{l}$ to be the expected smoothed loss where we flip each label with probability $p$:

$$\bar{l}(x) = pl(-x) + (1 - p)l(x). \tag{128}$$

| | $l(x)$ | $c = \arg\min_x \bar{l}(x)$ | $\sigma^2 = E[\epsilon^2]$ | $\alpha = \bar{l}''(c)$ |
|---|---|---|---|---|
| Logistic Loss | $\log\left[1 + e^{-x}\right]$ | $\log\frac{1-p}{p}$ | $p(1-p)$ | $p(1-p)$ |
| Exponential Loss | $e^{-x}$ | $\frac{1}{2}\log\frac{1-p}{p}$ | $1$ | $2\sqrt{p(1-p)}$ |
| Square Loss | $\frac{1}{2}(x-1)^2$ | $1 - 2p$ | $4p(1-p)$ | $1$ |

Table 1: Values of $l(x)$, $c$, $\sigma^2$, $\alpha$ for different binary classification loss functions

We make the following mild assumption on the smoothed loss $\bar{l}$ which is explicitly verified for the logistic loss, exponential loss, and square loss in Appendix E.2:

**Assumption 4** (Quadratic Approximation). *If $c \in \mathbb{R}$ is the unique global minimizer of $\bar{l}$, there exist constants $\epsilon_Q > 0, \nu > 0$ such that if $\bar{l}(x) \leq \epsilon_Q$ then,*

$$(x - c)^2 \leq \nu(\bar{l}(x) - \bar{l}(c)). \tag{129}$$

*In addition, we assume that $\bar{l}', \bar{l}''$ are $\rho_l, \kappa_l$ Lipschitz respectively restricted to the set $\{x : \bar{l}(x) \leq \epsilon_Q\}$.*

We verify Assumption 4 in Appendix E.2 for logistic loss, exponential loss, and square loss. Then we define the per-sample loss and the sample loss as:

$$\ell_i(\theta) = \bar{l}(y_i f_i(\theta)) - \bar{l}(c) \qquad \text{and} \qquad L(\theta) = \frac{1}{n}\sum_{i=1}^{n}\ell_i(\theta). \tag{130}$$

We will follow Algorithm 2:

---
**Algorithm 2:** SGD with Label Smoothing

---
**Input:** $\theta_0$, step size $\eta$, smoothing constant $p$, batch size $B$, steps $T$, loss function $l$
**for** $k = 0$ *to* $T - 1$ **do**

    Sample batch $\mathcal{B}^{(k)} \sim [n]^B$ uniformly and sample $\sigma_i^{(k)} = 1, -1$ with probability $1 - p, p$
    respectively for $i \in \mathcal{B}^{(k)}$.
    Let $\hat{\ell}_i^{(k)}(\theta) = l[\sigma_i^{(k)} y_i f_i(\theta)]$ and $\hat{L}^{(k)} = \frac{1}{B}\sum_{i \in \mathcal{B}^{(k)}} \hat{\ell}_i^{(k)}$.
    $\theta_{k+1} \leftarrow \theta_k - \eta\nabla\hat{L}^{(k)}(\theta_k)$

**end**

---

Now note that the noise per sample from label smoothing at a zero loss global minimizer $\theta^*$ can be written as

$$\nabla\hat{\ell}_i^{(k)}(\theta^*) - \nabla\ell_i(\theta^*) = \epsilon\nabla f_i(\theta^*) \tag{131}$$

where

$$\epsilon = \begin{cases} p(l'(c) + l'(-c)) & \text{with probability } 1 - p \\ -(1-p)(l'(c) + l'(-c)) & \text{with probability } p \end{cases} \tag{132}$$

so $E[\epsilon] = 0$ and

$$\sigma^2 = E[\epsilon^2] = p(1-p)(l'(c) + l'(-c))^2, \tag{133}$$

which will determine the strength of the regularization in Theorem 2. Finally, in order to study the local behavior around $c$ we define $\alpha = \bar{l}''(c) > 0$ by Assumption 4. Corresponding values for $c, \sigma^2, \alpha$ for logistic loss, exponential loss, and square loss are given in Table 1.

As before we define:

$$R(\theta) = -\frac{1}{2\eta\alpha}\operatorname{tr}\log\left(1 - \frac{\eta}{2}\nabla^2 L(\theta)\right), \qquad \lambda = \frac{\eta\sigma^2}{B}, \qquad \tilde{L}(\theta) = L(\theta) + \lambda R(\theta). \tag{134}$$

Our main result is a version of Theorem 1:

**Theorem 2.** *Assume that $f$ satisfies Assumption 1, $\eta$ satisfies Assumption 2, $L$ satisfies Assumption 3 and $l$ satisfies Assumption 4. Let $\eta, B$ be chosen such that $\lambda := \frac{\eta\sigma^2}{B} = \tilde{\Theta}(\min(\epsilon^{2/\delta}, \gamma^2))$, and let $T = \tilde{\Theta}(\eta^{-1}\lambda^{-1-\delta}) = \text{poly}(\eta^{-1}, \gamma^{-1})$. Assume that $\theta$ is initialized within $O(\sqrt{\lambda^{1+\delta}})$ of some $\theta^*$ satisfying $L(\theta^*) = O(\lambda^{1+\delta})$. Then for any $\zeta \in (0,1)$, with probability at least $1 - \zeta$, if $\{\theta_k\}$ follows Algorithm 2 with parameters $\eta, \sigma, T$, there exists $k < T$ such that $\theta_k$ is an $(\epsilon, \gamma)$-stationary point of $\frac{1}{\lambda}\tilde{L}$.*

### E.1 Proof of Theorem 2

The proof of Theorem 2 is virtually identical to that of Theorem 1. First we make a few simplifications without loss of generality:

First note that if we scale $l$ by $\frac{1}{\alpha}$ and $\eta$ by $\alpha$ then the update in Algorithm 2 i remain constant. In addition, $\frac{1}{\lambda}\tilde{L} = \frac{1}{\lambda}L + R$ remains constant. Therefore it suffices to prove Theorem 2 in the special case when $\alpha = 1$.

Next note that without loss of generality we can replace each $f_i$ with $y_i f_i$ and set all of the true labels $y_i$ to 1. Therefore from now on we will simply speak of $f_i$.

Let $\{\tau_m\}$ be a sequence of coupling times and $\{\theta_m^*\}$ a sequence of reference points. Let $T_m = \sum_{j<m} \tau_m$. Then for $k \in [T_m, T_{m+1})$, if $L^{(k)}$ denotes true value of the loss on batch $\mathcal{B}^{(k)}$, we can decompose the loss as

$$\theta_{k+1} = \theta_k - \underbrace{\eta\nabla L(\theta_k)}_{\text{gradient descent}} - \underbrace{\eta[\nabla L^{(k)}(\theta_k) - \nabla L(\theta_k)]}_{\text{minibatch noise}} + \underbrace{\frac{\eta}{B}\sum_{i\in\mathcal{B}^{(k)}}\epsilon_i^{(k)}\nabla f_i(\theta_k)}_{\text{label noise}} \quad (135)$$

where

$$\epsilon_i^{(k)} = \begin{cases} -p[l'(f_i(\theta_k)) + l'(-f_i(\theta_k))] & \sigma_i^{(k)} = 1 \\ (1-p)[l'(f_i(\theta_k)) + l'(-f_i(\theta_k))] & \sigma_i^{(k)} = -1. \end{cases} \quad (136)$$

We define

$$\epsilon_k = \frac{\eta}{B}\sum_{i\in\mathcal{B}^{(k)}}\epsilon_i^{(k)}\nabla f_i(\theta_k) \qquad \text{and} \qquad m_k = \nabla L^{(k)}(\theta_k) - \nabla L(\theta_k).$$

We decompose $\epsilon_k = \epsilon_k^* + z_k$ where

$$\epsilon_k^* = \frac{\eta}{B}\sum_{i\in\mathcal{B}^{(k)}}\epsilon_i^{(k)*}\nabla f_i(\theta_m^*) \qquad \text{where} \qquad \epsilon_i^{(k)*} = \begin{cases} -p[l'(f_i(c)) + l'(-f_i(c))] & \sigma_i^{(k)} = 1 \\ (1-p)[l'(f_i(c)) + l'(-f_i(c))] & \sigma_i^{(k)} = -1 \end{cases} \quad (137)$$

and $z_k = \epsilon_k - \epsilon_k^*$. Note that $\epsilon_k^*$ has covariance $\eta\lambda G(\theta_m^*)$. We define $\xi_0 = 0$ and for $k \in [T_m, T_{m+1})$,

$$\xi_{k+1} = (I - \eta G(\theta_m^*))\xi_k + \epsilon_k^*. \quad (138)$$

Then we have the following version of Proposition 6:

**Proposition 12.** *Let $\mathscr{X} = \sqrt{\max\left(\frac{p}{1-p}, \frac{1-p}{p}\right) \cdot \frac{2\lambda d\iota}{\nu}}$. Then for any $t \geq 0$, with probability $1 - 2de^{-\iota}$, $\|\xi_t\| \leq \mathscr{X}$.*

*Proof.* Let $P = \max\left(\frac{p}{1-p}, \frac{1-p}{p}\right)$. Define the martingale sequence $X_j^{(k)}$ as in Proposition 6. I claim that $[X^{(k)}, X^{(k)}]_k \preceq \frac{n\lambda P}{\nu}I$. We will prove this by induction on $k$. The base case is trivial as

$X_0^{(0)} = 0$. Then,

$$[X^{(k+1)}, X^{(k+1)}]_{k+1} = [X^{(k+1)}, X^{(k+1)}]_k + \epsilon_k^*(\epsilon_k^*)^T \tag{139}$$

$$= (I - \eta G_k)[X^{(k)}, X^{(k)}]_k(I - \eta G_k) + \epsilon_k^*(\epsilon_k^*)^T \tag{140}$$

$$\preceq \frac{n\lambda P}{\nu} \left[ (I - \eta G_k)^2 + \eta \nu G_k \right] \tag{141}$$

$$\preceq \frac{n\lambda P}{\nu} \left[ I - G_k(2 - \eta G_k - \nu I) \right] \tag{142}$$

$$\preceq \frac{n\lambda P}{\nu} I. \tag{143}$$

Therefore by Corollary 1 we are done. $\qquad\square$

Define $\iota, \mathscr{D}, \mathscr{M}, \mathscr{T}, \mathscr{L}$ as in Lemma 1. Then we have the following local coupling lemma:

**Lemma 9.** *Assume $f$ satisfies Assumption 1, $\eta$ satisfies Assumption 2, and $l$ satisfies Assumption 4. Let $\Delta_m = \theta_{T_m} - \xi_{T_m} - \theta_m^*$ and assume that $\|\Delta_m\| \leq \mathscr{D}$ and $L(\theta_m^*) \leq \mathscr{L}$ for some $0 < \delta \leq 1/2$. Then for any $\tau_m \leq \mathscr{T}$ satisfying $\max_{k \in [T_m, T_{m+1})} \|\Phi_{k-T_m}(\theta_m^* + \Delta_m) - \theta_m^*\| \leq 8\mathscr{M}$, with probability at least $1 - 10d\tau_m e^{-\iota}$ we have simultaneously for all $k \in (T_m, T_{m+1}]$,*

$$\|\theta_k - \xi_k - \Phi_{k-T_m}(\theta_m^* + \Delta_m)\| \leq \mathscr{D}, \qquad \mathbb{E}[\xi_k] = 0, \qquad and \qquad \|\xi_k\| \leq \mathscr{X}.$$

The proof of Lemma 9 follows directly from the following decompositions:

**Proposition 13.** *Let $\nabla^2 L = \nabla^2 L(\theta_m^*)$, $\nabla^3 L = \nabla^3 L(\theta_m^*)$, $G = G(\theta_m^*)$, $f_i = f_i(\theta_m^*)$, $g_i = \nabla f_i(\theta_m^*)$, $H_i = \nabla^2 f_i(\theta_m^*)$. Then,*

$$\nabla^2 L = G + O(\sqrt{\mathscr{L}}) \qquad and \qquad \frac{1}{2}\nabla^3 L(S) = \frac{1}{n}\sum_i H_i vv^T g_i + g_i O(\|v\|^2) + O(\|v\|^2\sqrt{\mathscr{L}}). \tag{144}$$

*Proof.* First, note that

$$\nabla^2 L = \frac{1}{n}\sum_i l''(f_i)g_i g_i^T + l'(f_i)H_i \tag{145}$$

$$= G + \ell\sqrt{\frac{1}{n}\sum_i [l''(f_i) - l''(c)]^2} + \rho_f\sqrt{\frac{1}{n}\sum_i [l'(f_i)]^2} \tag{146}$$

$$= G + O(\sqrt{\mathscr{L}}) \tag{147}$$

by Assumption 4. Next,

$$\frac{1}{2}\nabla^3 L(v, v) = \frac{1}{2n}\sum_i 2l''(f_i)H_i vv^T g_i + g_i[l'''(f_i)(g_i^T v)^2 + l''(f_i)v^T H_i v] + O(l'(f_i)) \tag{148}$$

$$= \frac{1}{n}\sum_i H_i vv^T g_i + g_i O(\|v\|^2) + O(\|v\|^2\sqrt{\mathscr{L}}). \tag{149}$$

$$\square$$

These are the exact same decompositions used Proposition 3 and Proposition 4, so Lemma 9 immediately follows. In addition, as we never used the exact value of the constant in $\mathscr{X}$ in the proof of Theorem 1, the analysis there applies directly as well showing that we converge to an $(\epsilon, \gamma)$-stationary point and proving Theorem 2.

### E.2 Verifying Assumption 4

We verify Assumption 4 for the logistic loss, the exponential loss, and the square loss and derive the corresponding values of $c, \sigma^2$ found in Table 1.

### E.2.1 Logistic Loss

For logistic loss, we let $l(x) = \log(1 + e^{-x})$, and $\bar{l}(x) = pl(-x) + (1-p)l(x)$. Then

$$\bar{l}'(x) = \frac{pe^x - (1-p)}{1 + e^x} \tag{150}$$

which is negative when $x < \log \frac{1-p}{p}$ and positive when $x > \log \frac{1-p}{p}$ so it is minimized at $c = \log \frac{1-p}{p}$. To show the quadratic approximation holds at $c$, it suffices to show that $\bar{l}'''(x)$ is bounded. We have $\bar{l}''(x) = \frac{e^x}{(1+e^x)^2}$ and

$$\bar{l}'''(x) = \frac{e^x(1 - e^x)}{(1 + e^x)^3} < \frac{1}{4} \tag{151}$$

so we are done. Finally, to calculate the strength of the noise at $c$ we have

$$\sigma^2 = p(1-p)(l'(c) + l'(-c))^2 = p(1-p)(-p + p - 1)^2 = p(1-p). \tag{152}$$

### E.2.2 Exponential Loss

We have $l(x) = e^{-x}$ and $\bar{l}(x) = pl(-x) + (1-p)l(x)$. Then,

$$\bar{l}'(x) = pe^x - (1-p)e^{-x} \tag{153}$$

which is negative when $x < \frac{1}{2} \log \frac{1-p}{p}$ and positive when $x > \frac{1}{2} \log \frac{1-p}{p}$ so it is minimized at $c = \frac{1}{2} \log \frac{1-p}{p}$. Then we can compute

$$\bar{l}(c + x) = 2\sqrt{p(1-p)} \cosh(x) \geq 2\sqrt{p(1-p)} + \sqrt{p(1-p)}x^2 = L^* + \sqrt{p(1-p)}x^2 \tag{154}$$

because $\cosh x \geq 1 + \frac{x^2}{2}$. Finally to compute the strength of the noise we have

$$\sigma^2 = p(1-p)(l'(c) + l'(-c))^2 = p(1-p)\left(-\sqrt{\frac{p}{1-p}} - \sqrt{\frac{1-p}{p}}\right)^2 = 1. \tag{155}$$

### E.2.3 Square Loss

We have $l(x) = \frac{1}{2}(1-x)^2$ and $\bar{l}(x) = pl(-x) + (1-p)l(x)$. Then,

$$\bar{l}(x) = \frac{1}{2}[p(1+x)^2 + (1-p)(1-x)^2] = \frac{1}{2}[x^2 + x(4p - 2) + 1] \tag{156}$$

which is a quadratic minimized at $c = 1 - 2p$. The quadratic approximation trivially holds and the strength of the noise is:

$$\sigma^2 = p(1-p)(l'(c) + l'(-c))^2 = p(1-p)(-2p - 2(1-p))^2 = 4p(1-p) \tag{157}$$

## F  Arbitrary Noise

### F.1  Proof of Proposition 5

We follow the proof of Lemma 2. First, let $\epsilon_k = \sqrt{\eta\lambda}\Sigma^{1/2}(\theta_k)x_k$ with $x_k \sim N(0, I)$ and define $\epsilon_k^* = \sqrt{\eta\lambda}\Sigma^{1/2}(\theta^*)x_k$ and $z_k = \epsilon_k - \epsilon_k^*$. Let $H = \nabla^2 L(\theta^*)$, $\Sigma = \Sigma(\theta^*)$, and $\nabla R_S = \nabla R_S(\theta^*)$. Let $\alpha$ be the smallest nonzero eigenvalue of $H$. Unlike in Lemma 1, we will omit the dependence on $\alpha$.

First we need to show $S$ exists. Consider the update

$$S \leftarrow (I - \eta H)S(I - \eta H) + \eta\lambda\Sigma(\theta^*)$$

Restricted to the span of $H$, this is a contraction so it must converge to a fixed point. In fact, we can write this fixed point in a basis of $H$ explicitly. Let $\{\lambda_i\}$ be the eigenvalues of $H$. The following computation will be performed in an eigenbasis of $H$. Then the above update is equivalent to:

$$S_{ij} = (1 - \eta\lambda_i)(1 - \eta\lambda_j)S_{ij} + \eta\lambda\Sigma_{ij}(\theta^*).$$

Therefore if $\lambda_i, \lambda_j \neq 0$ we can set

$$S_{ij} = \frac{\lambda \Sigma_{ij}(\theta^*)}{\lambda_i + \lambda_j - \eta \lambda_i \lambda_j}. \tag{158}$$

Otherwise we set $S_{ij} = 0$. Note that this is the unique solution restricted to $\mathrm{span}(H)$. Next, define the Ornstein-Uhlenbeck process $\xi$ as follows:

$$\xi_{k+1} = (I - \eta H)\xi_k + \epsilon_k^*.$$

Then note that

$$\xi_k = \sum_{j<k}(I - \eta H)^j \epsilon_{k-j}$$

so $\xi$ is Gaussian with covariance

$$\eta \lambda \sum_{j<k}(I - \eta H)^j \Sigma (I - \eta H)^j.$$

This is bounded by

$$C\eta\lambda \sum_{j<k}(I - \eta H)^j H (I - \eta H)^j \preceq C\lambda(2 - \eta H)^{-1} \preceq \frac{C\lambda}{\nu} I$$

so by Corollary 1, $\|\xi_k\| \leq \mathscr{X}$ with probability $1 - 2de^{-\iota}$. Define $v_k = \theta_k - \Phi_k(\theta_0)$ and $r_k = \theta_k - \xi_k - \Phi_k(\theta_0)$. We will prove by induction that $\|r_t\| \leq \mathscr{D}$ with probability at least $1 - 8dte^{-\iota}$. First, with probability $1 - 2de^{-\iota}$, $\|\xi_t\| \leq \mathscr{X}$. In addition, for $k \leq t$,

$$\|\theta_k - \theta^*\| \leq 9\mathscr{D} + \mathscr{X} = O(\mathscr{X}). \tag{159}$$

Therefore from the second order Taylor expansion:

$$r_{k+1} = (I - \eta H)r_k - \eta \left[\frac{1}{2}\nabla^3 L(\xi_k, \xi_k) - \lambda \nabla R_S\right] + z_k + O(\eta \mathscr{X}(\mathscr{D} + \mathscr{X}^2)).$$

Because $z_k$ is Gaussian with covariance bounded by $O(\eta\lambda\mathscr{X}^2)$ by the assumption that $\Sigma^{1/2}$ is Lipschitz, we have by the standard Gaussian tail bound that its contribution after summing is bounded by $\sqrt{\eta\lambda\mathscr{X}^2 k\iota}$ with probability at least $1 - 2de^{-\iota}$ so summing over $k$ gives

$$r_{t+1} = -\eta \sum_{k\leq t}(I - \eta H)^{t-k}\left[\frac{1}{2}\nabla^3 L(\xi_k, \xi_k) - \lambda \nabla R_S\right] + O(\sqrt{\eta\lambda t}\mathscr{X} + \eta t\mathscr{X}(\mathscr{D} + \mathscr{X}^2)).$$

Now denote $S_k = \xi_k \xi_k^T$. Then we need to bound

$$\eta \sum_{k\leq t}(I - \eta H)^{t-k}\nabla^3 L(S_k - S).$$

Let $D_k = S_k - S$. Then plugging this into the recurrence for $S_k$ gives

$$D_{k+1} = (I - \eta H)D_k(I - \eta H) + W_k + Z_k$$

where

$$W_k = (I - \eta H)\xi_k(\epsilon_k^*)^T + \epsilon_k^*(\xi_k)^T(I - \eta H) \qquad \text{and} \qquad Z_k = \epsilon_k^*(\epsilon_k^*)^T - \eta\lambda\Sigma.$$

Then,

$$D_k = (I - \eta H)^k S(I - \eta H)^k + \sum_{j<k}(I - \eta H)^{k-j-1}(W_j + Z_j)(I - \eta H)^{k-j-1}$$

so we need to bound

$$\eta \sum_{k\leq t}(I - \eta H)^{t-k}\nabla^3 L\left[(I - \eta H)^k S(I - \eta H)^k + \sum_{j<k}(I - \eta H)^{k-j-1}(W_j + Z_j)(I - \eta H)^{k-j-1}\right].$$

Because $S$ is in the span of $H$,

$$\left\| \eta \sum_{k<t} (I - \eta H)^{t-k} \nabla^3 L \left[ (I - \eta H)^k S (I - \eta H)^k \right] \right\| = O(\eta \lambda) \left\| \sum_{k \leq t} (I - \eta H)^k \Pi_H \right\| = O(\lambda/\alpha) = O(\lambda).$$

where $\Pi_H$ is the projection onto $H$. We switch the order of summation for the next two terms to get

$$\eta \sum_{j \leq t} \sum_{k=j+1}^{t} (I - \eta H)^{t-k} \nabla^3 L \left[ (I - \eta H)^{k-j-1} (W_j + Z_j)(I - \eta H)^{k-j-1} \right].$$

Note that conditioned on $\epsilon_l^*, l < j$, the $W_j$ part of the inner sum is Gaussian with variance bounded by $O(\eta \lambda \mathscr{X}^2)$ so by Lemma 16, with probability at least $1 - 2de^{-\iota}$, the contribution of $W$ is bounded by $O(\sqrt{\eta \lambda t \iota} \mathscr{X})$.

For the $Z$ term, we will define a truncation parameter $r$ to be chosen later. Then define $\bar{x}_j = x_j[\|x_j\| \leq r]$ where $x_j \sim N(0, I)$ is defined above. Define $\bar{X} = \mathbb{E}[\bar{x}_j \bar{x}_j^T]$. Then we can decompose the $Z$ term into:

$$\eta^2 \lambda \sum_{j \leq t} \sum_{k=j+1}^{t} (I - \eta H)^{t-k} \nabla^3 L \left[ (I - \eta H)^{k-j-1} \Sigma^{1/2} \left( x_j x_j^T - \bar{x}_j \bar{x}_j^T \right) (\Sigma^{1/2})^T (I - \eta H)^{k-j-1} \right]$$

$$+ \eta^2 \lambda \sum_{j \leq t} \sum_{k=j+1}^{t} (I - \eta H)^{t-k} \nabla^3 L \left[ (I - \eta H)^{k-j-1} \Sigma^{1/2} \left( \bar{x}_j \bar{x}_j^T - \bar{X}_j \right) (\Sigma^{1/2})^T (I - \eta H)^{k-j-1} \right]$$

$$+ \eta^2 \lambda \sum_{j \leq t} \sum_{k=j+1}^{t} (I - \eta H)^{t-k} \nabla^3 L \left[ (I - \eta H)^{k-j-1} \Sigma^{1/2} \left( \bar{X} - I \right) (\Sigma^{1/2})^T (I - \eta H)^{k-j-1} \right].$$

With probability $1 - 2dte^{-r^2/2}$ we can assume that $x_j x_j^T = \bar{x}_j \bar{x}_j^T$ for all $j \leq t$ so the first term is zero. For the second term the inner sum is bounded by $O(r^2 \eta^{-1})$ and has variance bounded by $O(\eta^{-2})$ by the same arguments as above. Therefore by Bernstein's inequality, the whole term is bounded by $O(\eta \lambda \sqrt{t\iota} + r^2 \eta \lambda \iota)$ with probability $1 - 2de^{-\iota}$. Finally, to bound the third term note that

$$\left\| \bar{X} - I \right\|_F = \mathbb{E}[\|x_j\|^2 [\|x_j\| > r]] \leq \sqrt{\mathbb{E}[\|x_j\|^4] \Pr[\|x_j\| > r]} \leq (d+1)\sqrt{2d} e^{-r^2/4}.$$

Therefore the whole term is bounded by $O(\eta \lambda t e^{-r^2/4})$. Finally, pick $r = \sqrt{4\iota \log \mathscr{T}}$. Then the final bound is

$$r_{t+1} \leq O\left( \sqrt{\eta \mathscr{T}} \mathscr{X}^2 + \eta \mathscr{T} \mathscr{X} (\mathscr{D} + \mathscr{X}^2) \right)$$

$$= O\left( \frac{\lambda^{3/4} \iota^{1/4}}{c} \right)$$

$$\leq \mathscr{D}$$

for sufficiently large $c$. This completes the induction.

### F.2 SGD Cycling

Let $\theta = (x, y, z_1, z_2, z_3, z_4)$. We will define a set of functions $f_i$ as follows:

$$\begin{aligned}
f_1(\theta) &= (1-y)z_1 - 1, \quad f_2(\theta) = (1-y)z_1 + 1, \quad f_3(\theta) = (1+y)z_2 - 1, \quad f_4(\theta) = (1+y)z_2 + 1, \\
f_5(\theta) &= (1-x)z_3 - 1, \quad f_6(\theta) = (1-x)z_4 + 1, \quad f_7(\theta) = (1+x)z_4 - 1, \quad f_8(\theta) = (1+x)z_4 + 1, \\
f_9(\theta) &= (1-x)z_1, \qquad f_{10}(\theta) = (1+x)z_2, \qquad f_{11}(\theta) = (1+y)z_3, \qquad f_{12}(\theta) = (1-y)z_4, \\
f_{13}(\theta) &= x^2 + y^2 - 1
\end{aligned}$$

and we set all labels $y_i = 0$. Then we verify empirically that if we run minibatch SGD with the loss function $\ell_i(\theta) = \frac{1}{2}(f_i(\theta) - y_i)^2$ then $(x, y)$ cycles counter clockwise over the set $x^2 + y^2 = 1$:

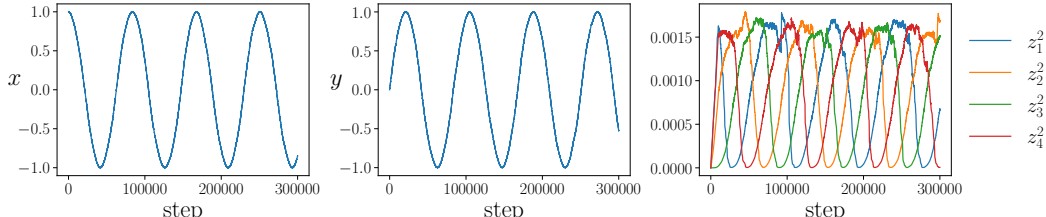

Figure 5: **Minibatch SGD can cycle.** We initialize at the point $\theta = (1, 0, 0, 0, 0, 0)$. The left column shows $x$ over time which follows a cosine curve. The middle column shows $y$ over time which follows a sine curve. Finally, the right column shows moving averages of $z_i^2$ for $i = 1, 2, 3, 4$, which periodically grow and shrink depending on the current values of $x, y$.

The intuition for the definition of $f$ above is as follows. When $x = 1$ and $y = 0$, due to the constraints from $f_9$ to $f_12$, only $z_1$ can grow to become nonzero. Then locally, $f_1 = z_1 - 1$ and $f_2 = z_1 + 1$ so this will cause oscillations in the $z_1$ direction, so $S$ will concentrate in the $z_1$ direction which will bias minibatch SGD towards decreasing the corresponding entry in $\nabla^2 L(\theta)$ which is proportional to $(1-x)^2 + 2(1-y)^2$, which means it will increase $y$. Similarly when $x = 0, y = 1$ there is a bias towards decreasing $x$, when $x = -1, y = 0$ there is a bias towards decreasing $y$, and when $x = 0, y = -1$ there is a bias towards increasing $x$. Each of these is handled by a different Ornstein Uhlenbeck process $z_i$. $f_{13}$ ensures that $\theta$ remains on $x^2 + y^2 = 1$ throughout this process. This cycling is a result of minimizing a rapidly changing potential and shows that the implicit bias of minibatch SGD cannot be recovered by coupling to a fixed potential.

# G    Weak Contraction Bounds and Additional Lemmas

Let $\{g_i\}_{i \in [n]}$ be a collection of $n$ vectors such that $\|g_i\|_2 \le \ell_f$ for all $i$ and let $G = \frac{1}{n} \sum_i g_i g_i^T$. Let the eigenvalues of $G$ be $\lambda_1, \ldots, \lambda_n$, and assume that $\eta$ satisfies Assumption 2. Then we have the following contraction bounds:

**Lemma 10.**
$$\|(I - \eta G)^\tau G\| \le \frac{1}{\eta \nu \tau} = O\left(\frac{1}{\eta \tau}\right) \tag{160}$$

**Lemma 11.**
$$\|(I - \eta G)^\tau g_i\| = O\left(\sqrt{\frac{1}{\eta \tau}}\right) \tag{161}$$

**Lemma 12.**
$$\sum_{k < \tau} \|(I - \eta G)^k g_{i_k}\| = O\left(\sqrt{\frac{\tau}{\eta}}\right) \tag{162}$$

**Lemma 13.**
$$\sum_{k < \tau} \|(I - \eta G)^k g_{i_k}\|^2 = O\left(\frac{1}{\eta}\right) \tag{163}$$

**Lemma 14.**
$$\sum_{k < \tau} \|(I - \eta G)^k g_{i_k}\| \|(I - \eta G)^k g_{j_k}\| = O\left(\frac{1}{\eta}\right) \tag{164}$$

**Lemma 15.**
$$\sum_{k < \tau} \|(I - \eta G)^k G\|_2 = O\left(\frac{1}{\eta}\right) \tag{165}$$

*Proof of Lemma 10.*

$$\|(I - \eta G)^\tau G\| = \max_i |1 - \eta\lambda_i|^\tau \lambda_i \le \max\left(\frac{\eta\lambda_i\tau}{\eta\tau}\exp(-\eta\lambda_i\tau), \ell(1-\nu)^\tau\right) \tag{166}$$

$$\le \max\left(\frac{1}{e\eta\tau}, \frac{1}{\eta\nu\tau}(\eta\ell)[\nu\tau]e^{-\nu\tau}\right) \tag{167}$$

$$\le \frac{1}{\eta\nu\tau} \tag{168}$$

$$= O\left(\frac{1}{\eta\tau}\right) \tag{169}$$

where we used that the function $xe^{-x} < \frac{1}{e}$ is bounded. $\qquad\square$

*Proof of Lemma 11.* Note that

$$\|(I - \eta G)^\tau g_i\|^2 = \text{tr}\left[(I - \eta G)^\tau g_i g_i^T (I - \eta G)^\tau\right] \tag{170}$$

$$\le n\,\text{tr}((I - \eta G)^{2\tau} G) \tag{171}$$

$$= n\sum_i \lambda_i (1 - \eta\lambda_i)^{2\tau} \tag{172}$$

$$\le n\sum_i \max\left(\lambda_i \exp(-2\eta\lambda_i\tau), \ell(1-\nu)^\tau\right) \tag{173}$$

$$= O\left(\frac{1}{\eta\tau}\right), \tag{174}$$

where we used the fact that the function $xe^{-x} \le \frac{1}{e}$ is bounded. $\qquad\square$

*Proof of Lemma 12.* Following the proof of Lemma 11,

$$\left(\sum_{k<\tau}\|(I - \eta G)^k g_{i_k}\|\right)^2 \le \tau\sum_{k<\tau}\|(I - \eta G)^k g_{i_k}\|^2 \tag{175}$$

$$\le n\tau\sum_{k<\tau}\sum_i \lambda_i(1 - \eta\lambda_i)^{2k} \tag{176}$$

$$= O\left(\frac{\tau}{\eta}\right) \tag{177}$$

$\qquad\square$

*Proof of Lemma 13.*

$$\sum_{k<\tau}\|(I - \eta G)^k g_{i_k}\|^2 \le \sum_{k<\tau}\text{tr}\left[(I - \eta G)^k g_{i_k} g_{i_k}^T (I - \eta G)^k\right] \tag{178}$$

$$\le n\sum_{k<\tau}\text{tr}\left[(I - \eta G)^{2k} G\right] \tag{179}$$

$$\le n\sum_{k<\tau}\sum_i \lambda_i(1 - \eta\lambda_i)^{2k} \tag{180}$$

$$= O\left(\frac{1}{\eta}\right) \tag{181}$$

$\qquad\square$

*Proof of Lemma 14.*

$$\sum_{k<\tau} \|(I - \eta G)^k g_{i_k}\| \|(I - \eta G)^k g_{j_k}\| \tag{182}$$

$$\leq \left[\sum_{k<\tau} \|(I - \eta G)^k g_{i_k}\|^2\right]^{1/2} \left[\sum_{k<\tau} \|(I - \eta G)^k g_{j_k}\|^2\right]^{1/2} \tag{183}$$

$$= O(1/\eta) \tag{184}$$

by Lemma 13. $\square$

*Proof of Lemma 15.*

$$\sum_{k<\tau} \|(I - \eta G)^k G\| \leq \sum_{k\leq\tau} \sum_i (I - \eta \lambda_i)^k \lambda_i \tag{185}$$

$$= O\left(\frac{1}{\eta}\right). \tag{186}$$

$\square$

The following concentration inequality is from Jin et al. [15]:

**Lemma 16** (Hoeffding-type inequality for norm-subGaussian vectors). *Given $X_1, \ldots, X_n \in \mathbb{R}^d$ and corresponding filtrations $\mathcal{F}_i = \sigma(X_1, \ldots, X_n)$ for $i \in [n]$ such that for some fixed $\sigma_1, \ldots, \sigma_n$:*

$$\mathbb{E}[X_i|\mathcal{F}_{i-1}] = 0, \mathbb{P}[\|X_i\| \geq t|\mathcal{F}_{i-1}] \leq 2e^{-\frac{t^2}{2\sigma_i^2}},$$

*we have that for any $\iota > 0$ there exists an absolute constant $c$ such that with probability at least $1 - 2de^{-\iota}$,*

$$\left\|\sum_{i=1}^n X_i\right\| \leq c \cdot \sqrt{\sum_{i=1}^n \sigma_i^2 \cdot \iota}.$$

**Lemma 17.** *Assume that $L$ is analytic and $\theta$ is restricted to some compact set $\mathcal{D}$. Then there exist $\delta > 0, \mu > 0, \epsilon_{KL} > 0$ such that Assumption 3 is satisfied.*

*Proof.* It is known that there exist $\mu_\theta, \delta_\theta$ satisfying the KL-inequality in the neighborhood of any critical point $\theta$ of $L$, i.e. for every critical point $\theta$, there exists a neighborhood $U_\theta$ of $\theta$ such that for any $\theta' \in U_\theta$,

$$L(\theta') - L(\theta) \leq \mu_\theta \|\nabla L(\theta')\|^{1+\delta_\theta}.$$

Let $S = \{\theta \in \mathcal{D} : L(\theta) = L(\theta^*)\}$ for any global minimizer $\theta^*$. For every global min $\theta \in S$, let $U_\theta$ be a neighborhood of $\theta$ such that the KL inequality holds with constants $\mu_\theta, \delta_\theta$. Because $\mathcal{D}$ is compact and $S$ is closed, $S$ is compact and there must exist some $\theta_1, \ldots, \theta_n$ such that $S \subset \bigcup_{i\in[k]} U_{\theta_i}$. Let $\delta = \min_i \delta_{\theta_i}$. Then for all $i$, there must exist some $\mu_i$ such that $\mu_i, \delta$ satisfies the KL inequality and let $\mu = \max_i \mu_i$. Finally, let $U = \bigcup_i U_{\theta_i}$ which is an open set containing $S$. Then $\mathcal{D} \setminus U$ is a compact set and therefore $L$ must achieve a minimum $\epsilon_{KL}$ on this set. Note that $\epsilon_{KL} > 0$ as $S \subset U$. Then if $L(\theta) \leq \epsilon_{KL}$, $\theta \in U$ so $\mu, \delta$ satisfy the KL inequality at $\theta$. $\square$

## H  Extension to SGD with Momentum

We now prove Lemma 4. We will copy all of the notation from Section 3.1. As before we define $v_k = \theta_k - \Phi_k(\theta^*)$. Define $\xi$ by $\xi_0 = 0$ and

$$\xi_{k+1} = (I - \eta G)\xi_k + \epsilon_k^* + \beta(\xi_k - \xi_{k-1}).$$

We now define the following block matrices that will be crucial in our analysis:

$$A = \begin{bmatrix} I - \eta G + \beta I & -\beta I \\ 1 & 0 \end{bmatrix} \quad \text{and} \quad J = \begin{bmatrix} I \\ 0 \end{bmatrix} \quad \text{and} \quad B_j = J^T A^j J.$$

Then we are ready to prove the following proposition:

**Proposition 14.** *With probability* $1 - 2de^{-\iota}$, $\|\xi_k\| \leq \mathscr{X}$.

*Proof.* Define $\bar{\xi}_k = \begin{pmatrix} \xi_k \\ \xi_{k-1} \end{pmatrix}$. Then the above can be written as:

$$\bar{\xi}_{k+1} = A\bar{\xi}_k + J\epsilon_k^*$$

Therefore by induction,

$$\bar{\xi}_k = \sum_{j<k} A^{k-j-1} J\epsilon_j^* \implies \xi_k = \sum_{j<k} B_{k-j-1}\epsilon_j^*.$$

The partial sums form a martingale and by Proposition 21, the quadratic covariation is bounded by

$$(1-\beta)n\eta\lambda \sum_{j=0}^{\infty} B_j G B_j^T \preceq \frac{n\lambda}{\nu}\Pi_G$$

so by Corollary 1 we are done. $\qquad\square$

We will prove Lemma 4 by induction on $t$. Assume that $\|r_k\| \leq \mathscr{D}$ for $k \leq t$. First, we have the following version of Proposition 3:

**Proposition 15.** *Let* $\bar{r}_k = \begin{pmatrix} r_k \\ r_{k-1} \end{pmatrix}$. *Then,*

$$\bar{r}_{k+1} = A\bar{r}_k + J\left(-\eta\left[\frac{1}{2}\nabla^3 L(\xi_k, \xi_k) - \lambda\nabla R\right] + m_k + z_k + O(\eta\mathscr{X}(\sqrt{\mathscr{L}} + \mathscr{M} + \mathscr{X}^2))\right)$$

*Proof.* As before we have that

$$v_{k+1} = (I - \eta G)v_k - \eta\left[\frac{1}{2}\nabla^3 L(\xi_k, \xi_k) - \lambda\nabla R\right] + \epsilon_k^* + m_k + z_k$$
$$+ O(\eta\mathscr{X}(\sqrt{\mathscr{L}} + \mathscr{M} + \mathscr{X}^2)) + \beta(v_k - v_{k-1})$$

and subtracting the definition of $\xi_k$ proves the top block of the proposition. The bottom block is equivalent to the identity $r_k = r_k$. $\qquad\square$

**Proposition 16.**
$$r_{t+1} = -\eta\sum_{k\leq t} B_{t-k}\left[\frac{1}{2}\nabla^3 L(\xi_k, \xi_k) - \lambda\nabla R\right] + O\left(\sqrt{\eta\lambda t}\left(\sqrt{\mathscr{L}} + \mathscr{X}\right) + \eta t\mathscr{X}\left(\sqrt{\mathscr{L}} + \mathscr{M} + \mathscr{X}^2\right)\right).$$

*Proof.* We have from the previous proposition that

$$\bar{r}_{t+1} = \sum_{k\leq t} A^{t-k} J\left(-\eta\left[\frac{1}{2}\nabla^3 L(\xi_k, \xi_k) - \lambda\nabla R\right] + m_k + z_k + O(\eta\mathscr{X}(\sqrt{\mathscr{L}} + \mathscr{M} + \mathscr{X}^2))\right)$$

so

$$r_{t+1} = \sum_{k\leq t} B_{t-k}\left(-\eta\left[\frac{1}{2}\nabla^3 L(\xi_k, \xi_k) - \lambda\nabla R\right] + m_k + z_k + O(\eta\mathscr{X}(\sqrt{\mathscr{L}} + \mathscr{M} + \mathscr{X}^2))\right).$$

By Corollary 3, we know that $B_k$ is bounded by $\frac{1}{1-\beta}$ so the remainder term is bounded by $O(\eta t\mathscr{X}(\sqrt{\mathscr{L}} + \mathscr{M} + \mathscr{X}^2))$. Similarly, by the exact same concentration inequalities used in the proof of Proposition 4, we have that the contribution of the $m_k, z_k$ terms is at most $O\left(\sqrt{\eta\lambda t}\left(\sqrt{\mathscr{L}} + \mathscr{X}\right)\right)$ which completes the proof. $\qquad\square$

**Proposition 17.**
$$\eta\sum_{k\leq t} B_{t-k}\left[\frac{1}{2}\nabla^3 L(\xi_k, \xi_k) - \lambda\nabla R\right] = O\left(\sqrt{\eta t}\mathscr{X}^2 + \eta t\mathscr{X}\sqrt{\mathscr{L}}\right).$$

*Proof.* As in the proof of Proposition 4, we define

$$S^* = \lambda \left(2 - \frac{\eta}{1+\beta}\nabla^2 L\right)^{-1}, \qquad \bar{S} = \lambda \left(2 - \frac{\eta}{1+\beta}G\right)^{-1}, \qquad \text{and} \qquad S_k = \xi_k \xi_k^T. \quad (187)$$

Then note that $\nabla R = \frac{1}{2}\nabla^3 L(S^*)$ so it suffices to bound

$$\eta \sum_{k \leq t} B_{t-k} \nabla^3 L(S_k - S^*).$$

As before we can decompose this as

$$\eta \sum_{k \leq t} B_{t-k} \nabla^3 L(S_k - \bar{S}) + \eta \sum_{k \leq t} B_{t-k} \nabla^3 L(\bar{S} - S^*).$$

We will begin by bounding the second term. Note that

$$\eta \sum_{k \leq t} B_{t-k} \nabla^3 L(\bar{S} - S^*) = O(\eta \| \bar{S} - S^* \|).$$

We can rewrite this as

$$S^* - \bar{S} = \lambda \left[ (2 - \eta \nabla^2 L)^{-1} \left( (2 - \eta G) - (2 - \eta \nabla^2 L) \right) (2 - \eta G)^{-1} \right] = O(\eta \lambda \sqrt{\mathscr{L}}) \quad (188)$$

so this difference contributes at most $O(\eta^2 \lambda t \sqrt{\mathscr{L}}) = O(\eta t \mathscr{X} \sqrt{\mathscr{L}})$. For the first term, let $D_k = S_k - \bar{S}$. We will decompose $\nabla^3 L$ as before to get

$$\frac{1}{n} \sum_{i=1}^{n} \eta \sum_{k \leq t} B_{t-k} \left[ H_i D_k g_i + \frac{1}{2} g_i \operatorname{tr}(D_k H_i) + O(\sqrt{\mathscr{L}} \mathscr{X}^2) \right]. \quad (189)$$

The third term can be bound by the triangle inequality by Corollary 3 to get $O(\eta t \sqrt{\mathscr{L}} \mathscr{X}^2)$. The second term can be bound by Proposition 22 to get $O(\sqrt{\eta t} \mathscr{X}^2)$.

The final remaining term is the first term. Define

$$\bar{S}' = \lambda \begin{bmatrix} \bar{S} & (I - \frac{\eta}{1+\beta}G)\bar{S} \\ (I - \frac{\eta}{1+\beta}G)\bar{S} & \bar{S} \end{bmatrix}.$$

From the proof of Proposition 21, we can see that $\bar{S}'$ satisfies

$$\bar{S}' = A\bar{S}'A^T + (1-\beta)\eta\lambda J G J^T.$$

We also have:

$$\bar{\xi}_{k+1} = A\bar{\xi}_k + J\epsilon_k^*$$

so

$$\bar{\xi}_{k+1}\bar{\xi}_{k+1}^T = A\bar{\xi}_k\bar{\xi}_k^T A^T + J\epsilon_k^*\bar{\xi}_k^T A^T + A\bar{\xi}_k(\epsilon_k^*)^T J^T + J\epsilon_k^*\epsilon_k^* J^T.$$

Let $D'_k = \bar{\xi}_k\bar{\xi}_k^T - \bar{S}'$. Then,

$$D'_{k+1} = AD'_k A^T + W_k + Z_k$$

where $W_k = J\epsilon_k^*\bar{\xi}_k^T A^T + A\bar{\xi}_k(\epsilon_k^*)^T J^T$ and $Z_k = J[\epsilon_k^*\epsilon_k^* - (1-\beta)\eta\lambda G]J^T$. Then,

$$D'_k = A^k \bar{S}' A^k + \sum_{j<k} A^{k-j-1}[W_j + Z_j](A^T)^{k-j-1}$$

so

$$D_k = J^T A^k \bar{S}' A^k J + \sum_{j<k} J^T A^{k-j-1}[W_j + Z_j](A^T)^{k-j-1} J.$$

Plugging this into the first term, which we have not yet bounded, we get

$$\frac{1}{n} \sum_{i=1}^{n} \eta \sum_{k \leq t} B_{t-k} H_i \left[ J^T A^k \bar{S}' A^k J + \sum_{j<k} J^T A^{k-j-1}[W_j + Z_j](A^T)^{k-j-1} J \right] g_i.$$

For the first term in this expression we can use Proposition 22 to bound it by $O(\sqrt{\eta t}\lambda) \leq O(\sqrt{\eta t}\mathcal{X}^2)$. Therefore we are just left with the second term. Changing the order of summation gives

$$\eta \frac{1}{n} \sum_{i=1}^{n} \sum_{j \leq t} \sum_{k=j+1}^{t} B_{t-k} H_i J^T A^{k-j-1}(W_j + Z_j)(A^T)^{k-j-1} J g_i. \tag{190}$$

Recall that $\epsilon_j^* = \frac{\eta}{B} \sum_{l \in \mathcal{B}^{(j)}} \epsilon_l^{(j)} g_l$. First, isolating the inner sum for the $W$ term, we get

$$\sum_{k=j+1}^{t} B_{t-k} H_i J^T A^{k-j} \bar{\xi}_j (\epsilon_j^*)^T J^T A^{k-j-1} J g_i \tag{191}$$

$$+ \sum_{k=j+1}^{t} B_{t-k} H_i J^T A^{k-j-1} J \epsilon_j^* \bar{\xi}_j^T A^{k-j} J g_i.$$

$$= \frac{\eta}{B} \sum_{l \in \mathcal{B}^{(j)}} \epsilon_l^{(j)} \Big[ \sum_{k=j+1}^{t} B_{t-k} H_i J^T A^{k-j} \bar{\xi}_j g_l^T B_{k-j-1} g_i \tag{192}$$

$$+ \sum_{k=j+1}^{t} B_{t-k} H_i B_{k-j-1} g_l \bar{\xi}_j^T A^{k-j} J g_i \Big].$$

The inner sums are bounded by $O(\mathcal{X} \eta^{-1})$ by Proposition 24. Therefore by Lemma 5, with probability at least $1 - 2de^{-\iota}$, the contribution of the $W$ term in Equation (60) is at most $O(\sqrt{\eta \lambda k \iota} \mathcal{X}) = O(\sqrt{\eta k} \mathcal{X}^2)$. The final remaining term to bound is the $Z$ term in (60). We can write the inner sum as

$$\frac{\eta \lambda(1-\beta)}{B^2} \sum_{k=j+1}^{t} B_{t-k} H_i J^T A^{k-j-1} J \left( \frac{1}{\sigma^2} \sum_{l_1,l_2 \in \mathcal{B}^{(k)}} \epsilon_{l_1}^{(j)} \epsilon_{l_2}^{(j)} g_{l_1} g_{l_2}^T - G \right) B_{k-j-1} g_i \tag{193}$$

which by Proposition 24 is bounded by $O(\lambda)$. Therefore by Lemma 5, with probability at least $1 - 2de^{-\iota}$, the full contribution of $Z$ is $O(\eta \lambda \sqrt{t\iota}) = O(\sqrt{\eta t} \mathcal{X}^2)$. □

Putting all of these bounds together we get with probability at least $1 - 10de^{-\iota}$,

$$\|r_{t+1}\| = O\left[ \sqrt{\eta \mathcal{T}} \mathcal{X} (\sqrt{\mathcal{L}} + \mathcal{X}) + \eta \mathcal{T} \mathcal{X} (\sqrt{\mathcal{L}} + \mathcal{M} + \mathcal{X}^2) \right]$$

$$= O\left( \frac{\lambda^{1/2+\delta/2} \iota}{\sqrt{c}} \right) \leq \mathcal{D}$$

for sufficiently large $c$ which completes the induction.

### H.1 Momentum Contraction Bounds

Let $u_i, \lambda_i$ be the eigenvectors and eigenvalues of $G$. Consider the basis $\bar{U}$ of $\mathbb{R}^{2d}$: $[u_1, 0], [0, u_1], \dots, [u_d, 0], [0, u_d]$. Then in this basis, $A, J$ are block diagonal matrix with $2 \times 2$ and $2 \times 1$ diagonal blocks:

$$A_i = \begin{bmatrix} 1 - \eta \lambda_i + \beta & -\beta \\ 1 & 0 \end{bmatrix} \quad \text{and} \quad J_i = \begin{bmatrix} 1 \\ 0 \end{bmatrix}.$$

Let the eigenvalues of $A_i$ be $a_i, b_i$ so

$$a_i = \frac{1}{2}\left(1 - \eta\lambda_i + \beta + \sqrt{(1-\eta\lambda_i+\beta)^2 - 4\beta}\right) \quad b_i = \frac{1}{2}\left(1 - \eta\lambda_i + \beta - \sqrt{(1-\eta\lambda_i+\beta)^2 - 4\beta}\right).$$

Note that these satisfy $a_i + b_i = 1 - \eta\lambda_i + \beta$ and $a_i b_i = \beta$.

**Proposition 18.** *If $\eta \in (0, \frac{2(1+\beta)}{\ell})$, then $\rho(A_i) \leq 1$. If $\lambda_i \neq 0$ then $\rho(A_i) < 1$.*

*Proof.* First, if $(1 - \eta\lambda_i + \beta)^2 - 4B \leq 0$ then $|a_i| = |b_i| = \sqrt{\beta} < 1$ so we are done. Otherwise, we can assume WLOG that $\eta\lambda_i < 1 + \beta$ because $\rho(A_i)$ remains fixed by the transformation $\eta\lambda_i \to 2(1+\beta) - \eta\lambda_i$. Then $a_i > b_i > 0$ so it suffices to show $a_i < 1$. Let $x = 1 - \eta\lambda_i + \beta$. Then,

$$\frac{x + \sqrt{x^2 - 4\beta}}{2} < 1 \iff \sqrt{x^2 - 4\beta} < 2 - x \iff x < 1 + \beta.$$

and similarly for $\leq$ in place of $<$ so we are done. $\square$

**Proposition 19.** *Let $s_k = \sum_{j<k} a_i^{k-j-1} b_i^j$. Then,*

$$A_i^k = \begin{bmatrix} s_{k+1} & -\beta s_k \\ s_k & -\beta s_{k-1} \end{bmatrix}.$$

*Proof.* We proceed by induction on $k$. The base case is clear as $s_2 = a_i + b_i = 1 - \eta\lambda_i + \beta$, $s_1 = 1$, and $s_0 = 0$. Now assume the result for some $k \geq 0$. Then,

$$A_i^{k+1} = \begin{bmatrix} s_{k+1} & -\beta s_k \\ s_k & -\beta s_{k-1} \end{bmatrix} \begin{bmatrix} a_i + b_i & -\beta \\ 1 & 0 \end{bmatrix} = \begin{bmatrix} s_{k+1} & -\beta s_k \\ s_k & -\beta s_{k-1} \end{bmatrix}$$

because $(a_i + b_i)s_k - \beta s_{k-1} = (a_i + b_i)s_k - a_i b_i s_{k-1} = s_{k+1}$. $\square$

**Proposition 20.**

$$\left| J_i^T A_i^k J_i \right| \leq \frac{1}{1 - \beta}.$$

*Proof.* From the above proposition,

$$\left| J_i^T A_i^k J_i \right| \leq \sup_k |s_{k+1}|.$$

Then for any $k$,

$$|s_{k+1}| = \left| \sum_{j \leq k} a_i^{k-j} b_i^j \right| \leq \sum_{j \leq k} |a_i|^{k-j} |b_i|^j \leq \sum_{j \leq k} |a_i|^j |b_i|^j = \sum_{j \leq k} \beta^j \leq \frac{1}{1 - \beta}.$$

where the second inequality follows from the rearrangement inequality as $\{|a_i|^{k-j}\}_j$ is an increasing sequence and $\{|b_i|^j\}_j$ is a decreasing sequence. $\square$

**Corollary 3.**

$$\|B_k\|_2 \leq \frac{1}{1 - \beta}.$$

**Proposition 21.**

$$\sum_{j=0}^{\infty} B_j G B_j^T = \frac{1}{\eta(1 - \beta)} \Pi_G \left( 2 - \frac{\eta}{(1 + \beta)} G \right)^{-1}.$$

*Proof.* Consider $\sum_{j=0}^{\infty} A^j J G J^T (A^T)^j$. We will rewrite this expression in the basis $\bar{U}$. Then the $i$th diagonal block will be equal to

$$\lambda_i \sum_{j=0}^{\infty} A_i^j J_i \lambda_i J_i^T (A_i^T)^j = \lambda_i \sum_{j=0}^{\infty} \begin{bmatrix} s_{j+1}^2 & s_j s_{j+1} \\ s_j s_{j+1} & s_j^2 \end{bmatrix}.$$

If $\lambda_i = 0$ then this term is $0$. Otherwise, we know that $|a_i|, |b_i| < 1$ so this infinite sum converges to some matrix $S = \begin{bmatrix} s_{11} & s_{12} \\ s_{21} & s_{22} \end{bmatrix}$. Then plugging this into the fixed point equation gives

$$S_i = A_i S_i A_i^T + J_i \lambda_i J_i^T$$

and solving this system entry wise for $s_{11}, s_{12}, s_{21}, s_{22}$ gives

$$S_i = \frac{1}{\eta(1 - \beta)} \begin{bmatrix} \frac{1}{2 - \frac{\eta}{1+\beta}\lambda_i} & \frac{1+\beta-\eta\lambda_i}{2(1+\beta)-\eta\lambda_i} \\ \frac{1+\beta-\eta\lambda_i}{2(1+\beta)-\eta\lambda_i} & \frac{1}{2 - \frac{\eta}{1+\beta}\lambda_i} \end{bmatrix}.$$

Converting back to the original basis gives the desired result. $\square$

**Proposition 22.**

$$\sum_{k<\tau} \|A^k J g_i\| = O\left(\sqrt{\frac{\tau}{\eta}}\right) \tag{194}$$

*Proof.* By Cauchy we have that

$$\left(\sum_{k<\tau} \|A^k J g_i\|\right)^2 \le \tau \sum_{k<\tau} \|A^k J g_i\|^2 \tag{195}$$

$$\le \tau \sum_{k<\tau} \text{tr}[A^k J G J^T (A^k)^T] \tag{196}$$

$$\le O\left(\frac{\tau}{\eta}\right) \tag{197}$$

by Proposition 21. □

**Proposition 23.**

$$\sum_{k<\tau} \|A^k J g_{i_k}\|^2 = O\left(\frac{1}{\eta}\right).$$

*Proof.*

$$\sum_{k<\tau} \|A^k J g_i\|^2 \le \tau \sum_{k<\tau} \text{tr}[A^k J G J^T (A^k)^T] \le O\left(\sqrt{\frac{1}{\eta}}\right).$$

□

**Proposition 24.**

$$\sum_{k<\tau} \|A^k J g_{i_k}\| \|A^k J g_{j_k}\| = O\left(\frac{1}{\eta}\right).$$

*Proof.*

$$\left(\sum_{k<\tau} \|A^k J g_{i_k}\| \|A^k J g_{j_k}\|\right)^2 \le \left(\sum_{k\le\tau} \|A^k J g_{i_k}\|^2\right)\left(\sum_{k\le\tau} \|A^k J g_{j_k}\|^2\right) = O(1/\eta^2)$$

by Proposition 23. □