# OpenReview forum: "Label Noise SGD Provably Prefers Flat Global Minimizers"
_NeurIPS.cc/2021/Conference — NeurIPS 2021 Poster_

### Official Review · Reviewer_n6Kd · 2021-07-17

**Rating:** 6
**Confidence:** 3

**Summary:**

## Post-rebuttal
I thank the authors for their clarifications. We had a long discussion about the technical details, and some aspects became more clear to me, but it did not lead to a resolution of the concern that the theory is somewhat weak. I remain concerned about the restrictive nature of the assumptions regarding the quality of the local minimizer. The authors argue that existing theory guarantees us convergence to such minimizes, but there seems to be a mismatch between the requirements of this work and the proved statements in the cited papers.
######

This paper studies the question of minimizing a smooth loss function with noise injection inside labels. The authors show that SGD with label noise converges to a stationary point of a certain regularized loss function, where the regularization depends on the amount of noise, batch size, and the stepsize. The paper improves the results of prior work, namely it does not need small stepsizes of the work by Blanc et al., and the model is much more general compared to the paper of HaoChen et al. I find the topic of the work to be quite interesting and worthy of investigation, although it is hard for me to call this paper very insightful.

The results have a lot of novelty but they are also somewhat incremental and many limitations are still present in this paper. I particularly criticize the following aspects of the results: restrictive assumptions, requirement for local initialization (close to a global solution), and decreasing effect of regularization when the target stationarity is small.

**Limitations And Societal Impact:**

Yes

**Main Review:**

My main concern about this work is that it does move towards a good answer but the results are still under restrictive assumptions. First of all, in contrast to standard guarantees on convergence of SGD, the theory additionally requires Lipschitzness of stochastic functions and Hessians in addition to the standard assumption on Lipschitz gradients, which are all not applicable to non-smooth deep networks. I understand that at least some assumptions are required and I could see this as a necessary requirement to keep the theory somewhat simple, but the function also needs to be KL for global minima and Theorem 1 assumes that initialization is close to a global minimum. This essentially eliminates the possibility of encountering a local minimum by assumptions rather than by analysis, and I find this unrealistic to apply the obtained results to practical scenarios such as training neural networks. I understand the authors' argument that it is fine to assume local initialization since overparameterization helps to achieve almost zero loss, but the resulting local theory is not that interesting when combined with local-KL assumption.

It is also disappointing to see that one needs to have smaller lambda for smaller epsilon, which, as far as I can see, also requires decreasing the stepsize. Ideally I'd hope to see a result that allows to leverage overparameterization and achieve guarantees for non-decreasing regularization. Otherwise, it is not even clear why we should want to eliminate the assumption on small stepsizes of Blanc et al.

I still give the paper a weak accept because I think that the results may lead us towards understanding generalization and the factors that contribute to it. However, I cannot recommend a higher score as so many limitations are currently present.

### Minor
I have difficulty with understanding the right column in Figure 3. The figure caption reads "the right column displays their correlation" while the y-label is "Test accuracy". Could the authors clarify what exactly is shown there?

**Time Spent Reviewing:**

5

---

> ### Author Response · Authors · 2021-08-10
> **Response to Reviewer n6Kd**
>
> We would like to thank the reviewer for your detailed and thoughtful review. We hope our responses below address your concerns and that you would consider raising your score. Please let us know if you have any further questions or if anything is still unclear.
>
> **Q: In contrast to standard guarantees on convergence of SGD, the theory additionally requires Lipschitzness of stochastic functions and Hessians in addition to the standard assumption on Lipschitz gradients, which are all not applicable to non-smooth deep networks.**
>
> While it is true that the standard SGD convergence proof only requires Lipschitz gradients, it is standard to make additional smoothness assumptions to get stronger convergence and implicit regularization results [1, 2, 3]. While such theorems do not directly apply to ReLU networks, we discuss in the limitations section (Appendix A) that such assumptions are easy to verify for other activation functions that perform similarly in practice including softplus and SeLU. In addition, with these activations we empirically find that these smoothness parameters are fairly benign.
>
> **Q: ​​The function also needs to be KL for global minima and Theorem 1 assumes that initialization is close to a global minimum. This essentially eliminates the possibility of encountering a local minimum by assumptions rather than by analysis, and I find this unrealistic to apply the obtained results to practical scenarios such as training neural networks.**
>
> For the KL assumption, Lemma 15 shows that the KL inequality holds for any analytic function on a compact set, although it does not give an explicit bound on $\delta$. This includes neural networks with any analytic activation such as softplus/sigmoid/tanh. Our result still proves convergence to a stationary point for any $\delta > 0$.
>
> The point of this paper is not to show convergence to a global min of the training loss $L$, but to show convergence on the regularized loss $L + \lambda R$. We do not assume the KL inequality on the regularized loss, which is our final metric of optimality.
>
> The assumption that we initialize near a global minimum is made to decouple the low-loss implicit regularization effect of label noise SGD from the ability of GD/SGD to optimize the non-convex loss landscape starting at high loss. The latter problem is well studied and is beyond the scope of this paper. Any global convergence result (for example that neural networks in the NTK regime reach 0 training loss exponentially fast [4,5]) can be easily combined with our analysis to get an end to end result from arbitrary initialization. We omit this first phase of training because the analysis is inherently model and data distribution specific and has been well studied.
>
> **Q: I understand the authors' argument that it is fine to assume local initialization since overparameterization helps to achieve almost zero loss, but the resulting local theory is not that interesting when combined with local-KL assumption.**
>
> We want to clarify that combining NTK initialization theory along with our convergence result does not result in a local theory. A good mental picture for label noise SGD is that it minimizes the regularizer $R(\theta)$ constrained to the manifold $L(\theta) = 0$. The movement on the manifold can be very large and is not local, even with NTK initialization.
>
> The KL assumption allows for high dimensional manifolds of global minimizers and does not imply that GD/SGD will converge to a fixed solution. In particular, **it is not necessarily true that a stationary point of the *regularized loss* exists in a neighborhood of the random initialization** and label noise is therefore an effective tool for escaping the NTK initialization. Examining the interplay between the NTK initialization and label noise is therefore nontrivial and an interesting avenue for future research.
>
> **Q: It is also disappointing to see that one needs to have smaller lambda for smaller epsilon, which, as far as I can see, also requires decreasing the stepsize. Ideally I'd hope to see a result that allows to leverage overparameterization and achieve guarantees for non-decreasing regularization. Otherwise, it is not even clear why we should want to eliminate the assumption on small stepsizes of Blanc et al.**
>
> You are correct that in order to decrease $\epsilon$ it is also necessary to decrease $\lambda$. However, small $\lambda$ (but large $\eta$) is desirable in deep learning [6]. Because $\lambda$ also depends on $\sigma^2,B$ this does not imply that the step size $\eta$ must also be small.
>
> It is easy to see that even at reasonable batch sizes used in practice, $\eta$ can be large while $\lambda$ is small. For example, our experiments used $\eta$ as large as $5$, but with a batch size of $256$ and a label smoothing parameter of $p = 0.2$ which leads to an effective $\lambda = 3\times 10^{-3}$ (see Appendix D for the extension to classification). We therefore believe that the regime in which $\eta$ is large but $\lambda$ is small accurately reflects the choice of hyperparameters used in practice.
>
> Finally, using large $\eta$ but small $\lambda$ gives a much faster convergence rate than directly decreasing $\eta$. We discuss in section 2.4 that by picking a large learning rate and taking $\lambda = \epsilon^4$ you get an $\epsilon^{-6}$ convergence rate for the normalized regularized problem. If you instead pick $\eta,\lambda$ to be of the same order, the convergence rate is a much slower $\epsilon^{-10}$.
>
> **Q: I have difficulty with understanding the right column in Figure 3. The figure caption reads "the right column displays their correlation" while the y-label is "Test accuracy". Could the authors clarify what exactly is shown there?**
>
> The plot displays the correlation between the trace of the Hessian on the $x$ axis, which is a proxy for $R(\theta)$, and the test accuracy on the $y$ axis. The solid dots in the right column represent the values at the end of training while the lines represent the values throughout the training process. We have modified the captions for Figures 3 and 4 for improved clarity.
>
> The point of the plot is to demonstrate that both at the end of training and throughout the training process, the trace of the Hessian is strongly correlated with the test accuracy. In particular, on a log plot, the relationship is almost linear.
>
> **References**
>
> [1] Lyu, Kaifeng, and Jian Li. "Gradient descent maximizes the margin of homogeneous neural networks." arXiv preprint arXiv:1906.05890 (2019).
>
> [2] Nacson, Mor Shpigel, et al. "Lexicographic and depth-sensitive margins in homogeneous and non-homogeneous deep models." International Conference on Machine Learning. PMLR, 2019.
>
> [3] Gunasekar, Suriya, et al. "Implicit bias of gradient descent on linear convolutional networks." arXiv preprint arXiv:1806.00468 (2018).
>
> [4] Soltanolkotabi, Mahdi, Adel Javanmard, and Jason D. Lee. "Theoretical insights into the optimization landscape of over-parameterized shallow neural networks." IEEE Transactions on Information Theory 65.2 (2018): 742-769.
>
> [5] Du, Simon, et al. "Gradient descent finds global minima of deep neural networks." International Conference on Machine Learning. PMLR, 2019.
>
> [6] Wei, Colin, et al. "Regularization matters: Generalization and optimization of neural nets vs their induced kernel." (2019).

---

> > ### Comment · Reviewer_n6Kd · 2021-08-11
> > **Thank you for the response**
> >
> > Thank you for the detailed responses to my questions. I hope the answers below explain my review if something was unclear.
> >
> > 1. *"We hope our responses below address your concerns and that you would consider raising your score. "*
> > As I explained at the end of my review, I gave the paper an *optimistic* score as I find the results unsatisfactory but on an interesting topic. Unless the authors are able to fix some of the limitations, I don't have sufficient reasons to increase the score even higher.
> > 2. *"such assumptions are easy to verify for other activation functions that perform similarly in practice including softplus and SeLU"*
> > I don't think your statement is correct, even with identity/linear activations the product of two matrices gives $f$ that does not satisfy the assumptions. Similarly, SiLU activation functions is sufficiently smooth itself, but the assumptions seem to immediately break once two or more linear activations are used.
> > 3. *"The assumption that we initialize near a global minimum is made to decouple the low-loss implicit regularization effect of label noise SGD from the ability of GD/SGD to optimize the non-convex loss landscape starting at high loss."*
> > My point is that this assumption in combination with local KL property (which is indeed not restrictive on its own, but it is restrictive in combination with local initialization) makes the results less interesting.
> > 4. *"Any global convergence result (for example that neural networks in the NTK regime reach 0 training loss exponentially fast [4,5]) can be easily combined with our analysis to get an end to end result from arbitrary initialization. We omit this first phase of training because the analysis is inherently model and data distribution specific and has been well studied."*
> > The results that you refer to hold for a training loop with fixed overparameterized objective. Why would they apply to the settings where we introduce random noise to the labels, which effectively gives a new objective at every iteration (and the expectation is not overparameterized because for the same input we may have samples with different outputs)?
> > 5. *"We want to clarify that combining NTK initialization theory along with our convergence result does not result in a local theory"*
> > By locality I mean small initial loss.
> > 6. *"which leads to an effective  $\lambda=3\cdot 10^{3}"*
> > And since $\epsilon$ is of order $\lambda^{\frac{\delta}{2}}$ and $\delta\le \frac{1}{2}$, we end up with $\epsilon$ of order $(3\cdot 10^{-3})^\frac{1}{4} \approx 0.23$? As far as I can see, $\lambda$ needs to be tiny to make $\epsilon$ somewhat small.
> > 7. *"The point of the plot is to demonstrate that both at the end of training and throughout the training process, the trace of the Hessian is strongly correlated with the test accuracy."*
> > Thank you for the clarification.

---

> > > ### Author Response · Authors · 2021-08-12
> > > **Response to Reviewer n6Kd**
> > >
> > > We appreciate the clarifications and have added some minor updates to our paper to better address your concerns.
> > >
> > > **I don't think your statement is correct, even with identity/linear activations the product of two matrices gives f that does not satisfy the assumptions. Similarly, SiLU activation functions is sufficiently smooth itself, but the assumptions seem to immediately break once two or more linear activations are used.**
> > >
> > > It is true that the Lipschitz assumption does not hold globally for linear networks; however, it does apply when the weights and inputs are bounded. These types of smoothness assumptions are common in the literature [1,2,3] and are strictly necessary for model-agnostic convergence results. In order to remove or weaken our smoothness assumptions, we would have to specialize our results to specific models which would be a topic for a separate paper.
> > >
> > > **The results that you refer to hold for a training loop with fixed overparameterized objective. Why would they apply to the settings where we introduce random noise to the labels, which effectively gives a new objective at every iteration (and the expectation is not overparameterized because for the same input we may have samples with different outputs)?**
> > >
> > > Label noise SGD does not change the objective function at every iteration. It computes a noisy gradient (eq 3) for the true loss $L(\theta)$, which is overparameterized. In particular, the NTK convergence proofs extend to SGD [4] and, in particular, label noise SGD.
> > >
> > > At a high level, these proofs first prove that the network locally satisfies the PL property and then use the standard global convergence analysis for the PL inequality. In particular, the PL inequality is known to imply convergence to approximate global minimizers for both GD and SGD [5,6].
> > >
> > > We apologize for not making the justification for the low loss assumption clear in the paper, and we will state and prove the global convergence proof for the NTK initialization and label noise SGD in section 2.4 and the appendix. This gives an end to end result for deep neural networks with the NTK initialization that does not require the low loss initialization assumption.
> > >
> > > **My point is that this assumption in combination with local KL property (which is indeed not restrictive on its own, but it is restrictive in combination with local initialization) makes the results less interesting.**
> > >
> > > Could you clarify why you find the combination of local KL and low loss initialization restrictive?
> > >
> > > We agree with your statement that the local KL property is not restrictive on its own. The only additional assumption we make is that GD/SGD is able to effectively optimize the non-convex loss landscape to reach an approximate global minimizer. This assumption is both necessary and general, as it can be verified independently for different architectures. In particular, this assumption has been explicitly verified for deep neural networks with NTK initialization for both GD and SGD and we will state and prove a version specific to label noise SGD in the paper.
> > >
> > > **​​As far as I can see, $\lambda$ needs to be tiny to make $\epsilon$ somewhat small.**
> > >
> > > We agree with your assessment; however, as we mentioned in our previous response, $\lambda \to 0$ is often optimal in deep learning (e.g. in [6]). This problem setup is also equivalent to finding an approximate KKT point for the constrained problem
> > >
> > > $$\min_\theta R(\theta) \qquad\text{such that}\qquad L(\theta) = 0,$$
> > >
> > > for which the exact value of $\lambda$ is not important. In particular, the approximate KKT problem for this constrained formulation is: Given $\epsilon,\varphi$, find $\lambda, \theta$ such that
> > >
> > > $$\left\|\frac{1}{\lambda} \nabla L(\theta) + \nabla R(\theta)\right\| \le \epsilon \qquad\text{and}\qquad L(\theta) \le \varphi.$$
> > >
> > > Theorem 1 shows that by carefully picking $\lambda$, SGD with label noise solves this constrained problem in $\epsilon^{-6}$ steps.
> > >
> > > We apologize for not making this connection clear in the paper and we have added a brief discussion of the relationship between our problem formulation and the approximate constrained KKT problem in section 2.3.
> > >
> > > **References**
> > >
> > > [1] Chizat, Lenaic, Edouard Oyallon, and Francis Bach. "On lazy training in differentiable programming." arXiv preprint arXiv:1812.07956 (2018).
> > >
> > > [2] Liu, Chaoyue, Libin Zhu, and Mikhail Belkin. "Loss landscapes and optimization in over-parameterized non-linear systems and neural networks." arXiv preprint arXiv:2003.00307 (2020).
> > >
> > > [3] Jin, Chi, et al. "How to escape saddle points efficiently." International Conference on Machine Learning. PMLR, 2017.
> > >
> > > [4] Chen, Zixiang, et al. "A generalized neural tangent kernel analysis for two-layer neural networks." arXiv preprint arXiv:2002.04026 (2020).
> > >
> > > [5] Lei, Yunwen, et al. "Stochastic gradient descent for nonconvex learning without bounded gradient assumptions." IEEE transactions on neural networks and learning systems 31.10 (2019): 4394-4400.
> > >
> > > [6] Karimi, Hamed, Julie Nutini, and Mark Schmidt. "Linear convergence of gradient and proximal-gradient methods under the polyak-łojasiewicz condition." Joint European Conference on Machine Learning and Knowledge Discovery in Databases. Springer, Cham, 2016.

---

> > > > ### Comment · Reviewer_n6Kd · 2021-08-13
> > > > **I agree it's still overparameterized**
> > > >
> > > > *"Label noise SGD does not change the objective function at every iteration. It computes a noisy gradient (eq 3) for the true loss $L(\theta)$, which is overparameterized."*
> > > > Right, I viewed the objective as the expectation with all labels, which is not overparameterized, but it's probably not the right way. Since the gradient of the objective with label noise is still unbiased, it seems that overparameterization still holds for the objective under consideration. Although I suspect that overparameterization would break if we considered label noise in classification, this is probably beyond the scope of this work.
> > > >
> > > > *"Could you clarify why you find the combination of local KL and low loss initialization restrictive?"*
> > > > For two reasons. Firstly, because it's not true that SGD with any random seed empirically converges to optimal loss. Even on the simplest datasets like Cifar-10, it's hard to go below $10^{-4}$ train loss. On larger datasets like imagenet, we do not reach zero training loss, and tricks like knowledge distillation with thousands of epochs are required to train a neural network (like ResNet) to a very good train/test loss. I agree that overparameterization/low-loss theory is the best assumption we have at the moment, but it does not imply that it's a good assumption.
> > > > Secondly, the considered combination of assumptions makes the considered problem look very smooth and local-minima- and saddle-point-free. And it's not clear to me if without these assumptions the claims of your work would still hold.

---

### Official Review · Reviewer_C2mG · 2021-07-18

**Rating:** 7
**Confidence:** 3

**Summary:**

This work investigates the implicit regularization effect of stochastic gradient descent (SGD) with noisy labels.  The authors show that SGD with label noise converges to a stationary point of a regularized loss function. The regularization weight can be controlled with the step size, the batch size, and the noise level of the labels.  The authors assume that the loss function to be minimized satisfies standard regularity conditions, e.g., lipschitness and smoothness,  the step size is not too large (relative to the lipschitzness constant), and that the loss is not too flat around any global minimizer.

Assuming that the SGD is initialized close to an approximate global minimizer of the loss $L$ satisfying the above properties, the authors show that SGD with noisy labels will converge to an approximate stationary point of the regularized loss $\widetilde{L}$ (they give the expression of the regularized loss as a function of the step size, batch size, and noise level).  Since the authors do not assume the step size to tend to $0$ as in [1] they find that the reguralization term interpolates between penalizing all eigenvalues of $\nabla^2 L$ equally that happens for small learning rates and penalizing larger eigenvalues more (happens for large learning rates).

**Limitations And Societal Impact:**

 The authors adequately addressed the limitations and potential negative societal impact of their work.

**Main Review:**

I think the result presented in this work is interesting and a good step towards better understanding of the implicit regularization effect of SGD.  Moreover, I found interesting the interplay between step size and regularization showed in this work (in [1] the step size had to be very small).  The paper is well-written, easy to follow in general, and the results are technically non-trivial and interesting.  I believe that this work meets the standards of NeurIPS and I recommend acceptance.

In Theorem 1 it is assumed that SGD is initialized close to some approximate minimizer of $L$.  Since, at a high level, it seems that initializing the SGD at the global minimizer of $L$ makes it harder for it to escape, is this assumption really necessary?

Since we have that if the KL condition holds for some $\delta$ it also holds for any $\delta'< \delta$, is $\delta$ supposed to be in $(0, 1/2]$ in Assumption 3? Why not simply have $\delta \in (0, 1]$?

[1]:G. Blanc, N. Gupta, G. Valiant, and P. Valiant.  Implicit regularization for deep neural networks driven by an ornstein-uhlenbeck like process.

**Time Spent Reviewing:**

8 hours

---

> ### Author Response · Authors · 2021-08-10
> **Response to Reviewer C2mG**
>
> We would like to thank the reviewer for your detailed and thoughtful review. Please let us know if you have any further questions or if anything is still unclear.
>
> **Q: In Theorem 1 it is assumed that SGD is initialized close to some approximate minimizer of $L$. Since, at a high level, it seems that initializing the SGD at the global minimizer of $L$ makes it harder for it to escape, is this assumption really necessary?**
>
> This is a good point and at a high level you are correct that starting at high loss (e.g. a random initialization) is better. However, we wanted to decouple the low-loss implicit regularization effect of label noise SGD from the ability of GD/SGD to optimize the non-convex loss landscape starting at high loss. Any result that guarantees convergence to an approximate global minimizer (e.g. NTK convergence results or global PL/KL assumptions) can be combined with our analysis for an end to end result.
>
> In particular, this will not affect the rate of convergence to a stationary point of the regularized loss because at a high-level the dynamics will be to quickly converge to a nearby global minimizer of the loss and then to begin to optimize for flatness. For simplicity we omitted this first step as it is model and data dependent and has been well studied in other papers.
>
> **Q: Since we have that if the KL condition holds for some $\delta$ it also holds for any $\delta'<\delta$, is $\delta$ supposed to be in $(0,½]$ in Assumption 3? Why not simply have $\delta \in (0,1]$?**
>
> This is mostly a matter of presentation. You are correct that our analysis supports larger values of $\delta \in (0,1]$, however the convergence rate does not improve with $\delta$ for $\delta \in [½,1]$. We could alternatively have written the assumption as $\delta \in (0,1]$ and then require $\lambda \le \min(\epsilon^{2/\delta},\epsilon^4,\gamma^2)$ in Theorem 1. However, as you correctly pointed out, if KL holds with $\delta = 1$ it also holds with $\delta = ½$ so we constrained $\delta \in (0,½]$ for simplicity.
>
> We are open to changing our convention or to adding a discussion of this choice in section 2.1.

---

> > ### Comment · Reviewer_C2mG · 2021-09-01
> > **Thank you for your response.**
> >
> > I would like to thank the authors for their response. After reading the other reviews and responses I still believe that this paper is above the bar for NeurIPS and vote for acceptance.
> >
> > ---
> >
> > Minor comment: I am fine with adding a remark about $\delta$.

---

### Official Review · Reviewer_oFfo · 2021-07-19

**Rating:** 5
**Confidence:** 2

**Summary:**

-The paper studies the behavior of SGD with label noise for over-parametrized models. The label noise setting is linked to an implicit regularization scheme, which is helpful to improve the search for "better" optimal as well as to understand the optimization landscape. The study focuses on the analysis for the "flat" regions. Theoretical analysis has been provided and empirical studies on ResNet18 and CIFAR10 are shown.

**Limitations And Societal Impact:**

-The main social and practical issue of deep learning schemes is associated with the misuse of optimization procedures. As discussed in Section 5, understanding the sharpness of the local optimal as well as the effect of learning rate is of interest in mitigating these issues, or at least provide some form of guidance to avoid less useful local optimal. Moreover, it will also be better to discuss more on the effect of how SGD behaves after reaching the locally flat region.


**Main Review:**

-The paper studies an interesting idea on optimization landscape with perturbation from noise. Specifically, the work presents the behavior of optimization for regions where the gradient is small, which is termed as "flat" region. The notion of flatness is described via \epsilon. The neighborhood of the "flat" region, within norm \gamma in parameter space, has been considered. The idea is to link SGD optimizing procedure for data with the presence of label noise to implicitly regularized learning objectives and studies the conditions where the algorithm reaches the "flat" region or its neighborhood. It is a concrete study that provides fruitful theoretical insights on the particular analysis. I believe this can be useful for the community to better understand the SGD procedure in various stances. However, there are some questions to be further addressed as well as related concerns (please see below).

-In the classification setting, label noise the class-conditional noise is coherent with the setting. However, in the regression setting in the main paper, label noise may not be the best term. As this setting is related to the denoise score matching (SM) scheme, how does the analysis related to denoise SM? Algorithm 1 is presenting noise in the uniform labels, which is confusing.
Moreover, in the regression setting, what is the distribution of noise? and how does the distribution or property of the noise (e.g. variance) affect the analysis or the results, (for instance, in terms of strength of implicit regularization)? Maybe linking to the behavior of denoise SM will be helpful for the community to further understand SGD with the type of regularization presented, as well as linking the important ideas for learning procedures.
In Theorem 1, the analysis shows that the optimization reaches the (\epsilon,\gamma)-stationary region. Is such a region always better than a sharp local optimal?

-I am not an expert in the area of this type of analysis. Overall, the idea and analysis are interesting. The proof is not carefully checked. The major significance and impact of the analysis are not clear enough. For people who may not be familiar with the particular analysis methodology, it is not easy to understand and appreciate the setting. It is also unclear that what goes next when SGD reaches (\epsilon,\gamma)-stationary region. Adding more explanations would help understand the full SGD procedure better.

-Some minor clarifications

In section 2.2, it will be useful to provide more explicit links between the trace log regularization scheme to the noisy label setting to support the implicit regularizer claim.
In definition2, more intuition can be provided for better understanding. It is an interesting metric to consider for the convergence analysis, but it will be easier to have both technical claims and interpretable explanations.


**Time Spent Reviewing:**

5hours

---

> ### Author Response · Authors · 2021-08-10
> **Response to Reviewer oFfo**
>
> We would like to thank the reviewer for your detailed and thoughtful review. We hope our responses below address your concerns and that you would consider raising your score. Please let us know if you have any further questions or if anything is still unclear.
>
> **Q: As this setting is related to the denoise score matching (SM) scheme, how does the analysis related to denoise SM?**
>
> We are not certain about which paper or scheme you are referring to. Are you referring to [1]?
>
> [1] Vincent, Pascal. "A connection between score matching and denoising autoencoders." Neural computation 23.7 (2011): 1661-1674.
>
> **Q:  Algorithm 1 is presenting noise in the uniform labels, which is confusing.**
>
> Could you elaborate on this (e.g. what is confusing)? Label Noise SGD (Algorithm 1) is a commonly studied algorithm both in theory and practice, e.g. [2], [3], [4], [5].
>
> **Q: Moreover, in the regression setting, what is the distribution of noise? and how does the distribution or property of the noise (e.g. variance) affect the analysis or the results, (for instance, in terms of strength of implicit regularization)?**
>
> The precise distribution of the label noise is given in Algorithm 1 and the resulting noise covariance for the gradient updates is $\eta \lambda G(\theta)$ where $G(\theta) = \frac{1}{n} \sum_{i=1}^n \nabla_\theta f(x_i,\theta) \nabla_\theta f(x_i,\theta)^T$, as discussed in the proof sketch (Section 3). The label noise variance directly controls $\lambda$, which is the effective regularization parameter in section 2.2 and Theorem 1. The effect of the covariance of the gradient noise for general SGD and the resulting implicit regularization is discussed in section 5.3.
>
> **Q: In Theorem 1, the analysis shows that the optimization reaches the (\epsilon,\gamma)-stationary region. Is such a region always better than a sharp local optimal?**
>
> Theorem 1 proves convergence to a stationary point of the regularized loss, which has a strong preference for flat regions of the loss landscape. There are many existing results that connect the flatness of a global minimizer with generalization error. We provide experiments in section 4 that demonstrate that SGD indeed finds flat minimizers and that these minimizers generalize well. Furthermore, we verify the common intuition that flat global minimizers generalize better by plotting the correlation between the Trace of the Hessian, which measures flatness, and the Test Accuracy (Figure 3, right column). The curves show that not only are the two strongly correlated at the end of training, but they remain correlated throughout the training process.
>
> Finally, in section 5.2 we provide a concrete generalization bound that upper bounds the generalization error by our regularizer $R(\theta)$.
>
> **References**
>
> [1] Vincent, Pascal. "A connection between score matching and denoising autoencoders." Neural computation 23.7 (2011): 1661-1674.
>
> [2] Szegedy, Christian, et al. "Rethinking the inception architecture for computer vision." Proceedings of the IEEE conference on computer vision and pattern recognition. 2016.
>
> [3] Shallue, Christopher J., et al. "Measuring the effects of data parallelism on neural network training." arXiv preprint arXiv:1811.03600 (2018).
>
> [4] HaoChen, Jeff Z., et al. "Shape matters: Understanding the implicit bias of the noise covariance." Conference on Learning Theory. PMLR, 2021.
>
> [5] Blanc, Guy, et al. "Implicit regularization for deep neural networks driven by an ornstein-uhlenbeck like process." Conference on learning theory. PMLR, 2020.

---

### Official Review · Reviewer_Yasg · 2021-07-20

**Rating:** 6
**Confidence:** 4

**Summary:**

This paper studies the global convergence of SGD/GD with label noise. Under suitable assumptions, it shows that if initialized properly and with suitable stepsize/batch size, label noise SGD/GD converges approximately to some stationary point of a regularized loss function. The results reveal a novel implicit regularization effect of label noise that approximately equals to adding an explicit regularizer. Notably, this regularization effect is more favorable than the implicit bias of vanilla SGD, as vanilla SGD cannot escape from poor global minima, while adding label noise regularizes certain Hessian norm and thus helps the iterates to escape and converge to flat/good minima (that achieve both small main loss and small regularization loss). Empirical results are provided to demonstrate this advantage of label noise regularization.

**Limitations And Societal Impact:**

See above.

**Main Review:**

# Pros:
+ The theoretic results provide intuition to understand the implicit regularization effect of adding label noise, which is shown empirically to be better than SGD implicit bias in some cases (like initialization from poor minima).
+ The introduced $(\epsilon, \gamma)$-stationary points that characterize a neighbor of a stationary point seem to be interesting for understanding the properties of a regularized objective.
+ Empirical observations about label noise are partly justified by the presented theory.

# Cons:
- l.50. Based on my knowledge, the initial learning rate is generally large for large batch SGD (e.g. linear scaling rule). Could you provide references here?
- Definition 2. The order of $\epsilon$ vs. $\gamma$ seems to be important, could you elaborate on this? What prevents us from setting $\gamma \approx \epsilon$ as happening in linear regression? I am trying to interpret Thm1 for linear regression but find several places are not fully clear. Discussions on how the theorem applies to linear regression could be helpful (e.g., how to setup the parameters in the thm when the considered model is linear regression).
- I am not sure how Thm1 helps in terms of justifying the title. The title claims label noise SGD prefers flat global "minimizers", but thm1 only shows convergence to approximate stationary point. Please elaborate.
- I am not sure how Thm1 helps in terms of justifying the experiments. Indeed $\theta^*$ could be a poor global minimizer of $L$ so that SGD cannot escape. But it remains unclear to me why $\theta_k$ in thm 1 is a good minimizer.
- I am not sure how Thm1 helps in terms of justifying the claims on the effect of initial large learning rate. Note that Thm1 requires $\eta/B \lesssim \gamma^2$, then why $\eta$ is considered to be "large" (e.g., l.276)? Are you allowing some annealing learning rate here?
- My next question is also about Thm1: it seems to me the whole considered iterates are in a neighbor of $\theta^*$ (please correct me if not). Then why thm1 could justify the "escaping" behavior of label noise SGD in the experiments?
- l.267. In the experiment a lower bound is adopted in order to compute the proposed regularizer. As we are considering a minimization problem, why does it make sense to minimize a lower bound (instead of an upper bound) of the objective?





# Small issues:
* l.125. $\theta^*$ -> $\theta$
* l.138. $\phi$ -> $\gamma$


# Overall
My current feeling for this paper is a weak reject, as the claims/title/experiments are not properly justified by the theorem. Authors' feedback is welcome to help me better understand this paper.




----

Post-rebuttal: after further discussions with the authors, my initial concerns are all solved. Therefore I would like to raise the score and recommend to accept the paper.

A suggestion for future revision is to provide examples to illustrate the order of the important quantities in the theorem, e.g., $\epsilon$ and $\gamma$ and others.

**Time Spent Reviewing:**

5

---

> ### Author Response · Authors · 2021-08-10
> **Response to Reviewer Yasg**
>
> We would like to thank the reviewer for your detailed and thoughtful review. We hope our responses below address your concerns and that you would consider raising your score. Please let us know if you have any further questions or if anything is still unclear.
>
> **Q: Discussions on how the theorem applies to linear regression could be helpful (e.g., how to setup the parameters in the thm when the considered model is linear regression).**
>
> SGD noise does not have an implicit regularization effect in linear regression. In particular, the Hessian is globally constant for linear regression so no minimizer is “flatter” than any other.
>
> It is possible to specialize these results to any non-linear model with non-constant Hessian, for example the quadratically parameterized linear regression model in [1] for which the implicit regularization of label noise is equivalent to weighted $\ell_1$ regularization. We will add a discussion in section 2.4 and a section in the Appendix that specializes Theorem 1 to the quadratically parameterized linear regression model to make the theorem easier to understand.
>
> **I am not sure how Thm1 helps in terms of justifying the title. The title claims label noise SGD prefers flat global "minimizers", but thm1 only shows convergence to approximate stationary point. Please elaborate.**
>
> Theorem 1 shows that label noise SGD converges to a global minimizer that is **also a stationary point of the regularized loss**. Since the regularizer is smaller whenever the global minimizer is flatter, this shows label noise SGD has a preference for flat minimizers. We note that you cannot hope in this generality to reach a global minimizer of the regularized loss without strict model and data assumptions.
>
> **I am not sure how Thm 1 helps in terms of justifying the experiments. Indeed $\theta^\star$ could be a poor global minimizer of $L$ so that SGD cannot escape. But it remains unclear to me why $\theta_k$ in thm 1 is a good minimizer.**
>
> Theorem 1 proves convergence to a stationary point of the regularized loss, which has a strong preference for flat regions of the loss landscape. There are many existing results that connect the flatness of a global minimizer with generalization error. We provide experiments in section 4 that demonstrate that SGD indeed finds flat minimizers and that these minimizers generalize well. Furthermore, we verify the common intuition that flat global minimizers generalize better by plotting the correlation between the Trace of the Hessian, which measures flatness, and the Test Accuracy (Figure 3, right column). The curves show that not only are the two strongly correlated at the end of training, but they remain correlated throughout the training process.
>
> Finally, in section 5.2 we provide a concrete generalization bound that upper bounds the generalization error by our regularizer $R(\theta)$.
>
> **Q: Note that Thm1 requires $\eta/B \le \gamma^2$. Then why η is considered to be "large" (e.g., l.276)?**
>
> Theorem 1 only needs $\frac{\eta\sigma^2}{B} < \gamma^2$. In particular, because the label noise is artificially added by the algorithm, $\eta$ can be made large by decreasing the noise level $\sigma^2/B$. It is also common in stochastic gradient algorithms to let the step size $\eta$ depend on the final desired accuracy and the strength of the noise.
>
> Our settings are also empirically practical. For example, our experiments used $\eta$ as large as $5$, but with a batch size of $256$ and a label smoothing parameter of $p = 0.2$ which leads to an effective $\lambda = 3\times 10^{-3}$ (see Appendix D for the extension to classification). We therefore believe that the regime in which $\eta$ is large but $\lambda$ is small accurately reflects the choice of hyperparameters used in practice.
>
> **My next question is also about Thm1: it seems to me the whole considered iterates are in a neighbor of $\theta^\star$ (please correct me if not). Then why thm1 could justify the "escaping" behavior of label noise SGD in the experiments?**
>
> We apologize that this was unclear. $\theta^*$ is a global minimizer of $L$ but not a stationary point of the regularized loss. Since the closest stationary point of $L+\lambda R$ can be arbitrarily far from $\theta^*$, Theorem 1 proves you can move large distances. The proof of Theorem 1 follows from many repeated applications of the local coupling lemma, Lemma 1. The result in Lemma 1 is local; however, because we are chaining together many such couplings, the total movement in Theorem 1 is $O(1)$, i.e. it is a global result and justifies the term “escaping.”
>
> In comparison, the previous work of Blanc et al. [2] only applies for $\eta \to 0$ and moves a distance of $\eta^{0.4}$ away from the starting point and therefore cannot reach a stationary point of the regularized loss.
>
> **Q: l.50. Based on my knowledge, the initial learning rate is generally large for large batch SGD (e.g. linear scaling rule). Could you provide references here?**
>
> While it is true in theory that in order to maintain a constant strength of implicit regularization one should scale the learning rate with the batch size, in practice this can lead to instabilities in training. For example, in [3,4,5] the authors found that following the linear scaling rule caused divergence with large batch sizes. In [5] the authors solve this problem with learning rate warmup, however this leads to poor performance at large batch sizes. One solution is to keep the learning rate fixed but add artificial noise (e.g. label noise) to emulate small batch training, rather than increasing the learning rate in an attempt to amplify the existing minibatch noise [4].
>
> **Q: l.267. In the experiment a lower bound is adopted in order to compute the proposed regularizer. As we are considering a minimization problem, why does it make sense to minimize a lower bound (instead of an upper bound) of the objective?**
>
> We would like to clarify a potential misunderstanding. In the experiments, we track the value of $\text{tr} \nabla^2 L$ as a lower bound throughout training but the algorithm is just SGD with label noise (Algorithm 1) and we never explicitly minimize or take the gradient of $\text{tr} \nabla^2 L$. We only use it to approximate the flatness of the minimizers found throughout training for the plots in Figure 3. The training process is still only implicitly regularized by $R(\theta)$.
>
> **References:**
>
> [1] HaoChen, Jeff Z., et al. "Shape matters: Understanding the implicit bias of the noise covariance." Conference on Learning Theory. PMLR, 2021.
>
> [2] Blanc, Guy, et al. "Implicit regularization for deep neural networks driven by an ornstein-uhlenbeck like process." Conference on learning theory. PMLR, 2020.
>
> [3] Golmant, Noah, et al. "On the computational inefficiency of large batch sizes for stochastic gradient descent." arXiv preprint arXiv:1811.12941 (2018).
>
> [4] Shallue, Christopher J., et al. "Measuring the effects of data parallelism on neural network training." arXiv preprint arXiv:1811.03600 (2018).
>
> [5] Goyal, Priya, et al. "Accurate, large minibatch sgd: Training imagenet in 1 hour." arXiv preprint arXiv:1706.02677 (2017).

---

> > ### Comment · Reviewer_Yasg · 2021-08-15
> > **Need clarifications**
> >
> > Thanks for the response.
> >
> > * I agree that linear model is not interesting here as there is no *flatter* minima. Since the $(\epsilon, \gamma)$-stationary point (for regularized loss) is a core notion in the results, I believe examples with an explicit setting of these quantities (including $\eta, \sigma, B, \delta$, etc) could be very helpful for readers to interpret the results. I am looking forward to see an interpretation for quadratically parameterized linear regression models.
> >
> > * As you have pointed out, Thm1 shows the iterates converge to a certain stationary point of the regularized loss. However, it is unclear to me why such point is also an approximate global minimizer of the (unregularized) loss. Without this, I am afraid Thm1 still cannot justify the title.
> >
> > * Intuitively, I kind of agree that $\theta_k$ in thm1 tends to be "flatter" than $\theta^*$. It makes sense to me using the flatness argument (where flat minimizer generalizes better) to justify the potentially good generalization of $\theta_k$. I would suggest the authors to clarify this in line 134 to avoid causing confusion, i.e., thm1 itself does not directly justify why $\theta_k$ generalizes better.
> >
> > * I agree with the authors that from the theorems one can decrease $\sigma$ to accommodate a large $\eta$.
> >
> > * I agree with the authors' comments on the "escaping" behavior of label-noise SGD. This will be appreciated as a novelty point in my evaluation.
> >
> > * I think we share the same veiwpoints on the large learning rate in the initial stage of large batch SGD. However the current line 50 is misleading and needs to be rephrased. In particular, the initial learning rate of large batch SGD should still be considered *large* comparing with that of small batch SGD.
> >
> > * Thanks for the clarification on $tr(\nabla^2 L)$. Measuring a lower bound of the "flatness" is a much smaller issue than minimizing a lower bound of an objective in my perspective.
> >
> >
> >
> > The biggest issue that prevents me from raising the score is: why thm1 shows a convergence to a global minimizer (of the unregularized loss)?
> >
> > Moreover, clarifying the order of the quantities in Thm1 (perhaps with an example) could be a bonus for the interpretability of the results.

---

> > > ### Author Response · Authors · 2021-08-18
> > > **Response to Reviewer Yasg**
> > >
> > > We appreciate the clarification regarding the title. Our proof shows that we converge to a neighborhood of a stationary point of the regularized loss that is also an approximate global minimizer ($L(\theta_m^*) \le \lambda^{1+\delta}$, see Lemma 3); however, we did not state this in Theorem 1. We have revised Theorem 1 to reflect that we find an approximate global minimizer of the training loss.
> > >
> > > We agree with your comments regarding lines 50 and 134 and we will also add a full discussion of the quadratic model, including explicit values for $\eta,\sigma,B,\delta$ to section 2.4 and the appendix.

---

> > > > ### Comment · Reviewer_Yasg · 2021-08-20
> > > > **Thanks for the response**
> > > >
> > > > Thanks for the very helpful discussions. My concerns are all well answered and I will raise the score.

---

### Decision · Program_Chairs · 2021-09-27

**Decision:**

Accept (Poster)

**Comment:**

This paper studies the convergence of SGD in the presence of label noise. The authors show that SGD with label noise converges to a stationary point of a certain regularized loss function, where the regularization depends on the amount of noise, batch size, and the stepsize. The assumptions on size of regularization and starting points are somewhat restrictive, but the paper increases the understanding of generalization for this set of problems.